

# Enhancing software defect prediction: a framework with improved feature selection and ensemble machine learning

Misbah Ali[1,*], Tehseen Mazhar[1,*], Amal Al-Rasheed[2], Tariq Shahzad[3], Yazeed Yasin Ghadi[4] and Muhammad Amir Khan[5]

[1] Department of Computer Science & Information Technology, Virtual University of Pakistan, Lahore, Pakistan
[2] Department of Information Systems, College of Computer and Information Sciences, Princess Nourah bint Abdulrahman University, Riyadh, Saudi Arabia
[3] Department of Computer Sciences, COMSATS University Islamabad, Sahiwal Campus, Sahiwal, Pakistan
[4] Department of Computer Science and Software Engineering, Al Ain University, Abu Dhabi, UAE
[5] School of Computing Sciences, College of Computing, Informatics and Mathematics, Universiti Teknologi MARA, Shah Alam, Selangor, Malaysia
[*] These authors contributed equally to this work.

## ABSTRACT

Effective software defect prediction is a crucial aspect of software quality assurance, enabling the identification of defective modules before the testing phase. This study aims to propose a comprehensive five-stage framework for software defect prediction, addressing the current challenges in the field. The first stage involves selecting a cleaned version of NASA's defect datasets, including CM1, JM1, MC2, MW1, PC1, PC3, and PC4, ensuring the data's integrity. In the second stage, a feature selection technique based on the genetic algorithm is applied to identify the optimal subset of features. In the third stage, three heterogeneous binary classifiers, namely random forest, support vector machine, and naïve Bayes, are implemented as base classifiers. Through iterative tuning, the classifiers are optimized to achieve the highest level of accuracy individually. In the fourth stage, an ensemble machine-learning technique known as voting is applied as a master classifier, leveraging the collective decision-making power of the base classifiers. The final stage evaluates the performance of the proposed framework using five widely recognized performance evaluation measures: precision, recall, accuracy, F-measure, and area under the curve. Experimental results demonstrate that the proposed framework outperforms state-of-the-art ensemble and base classifiers employed in software defect prediction and achieves a maximum accuracy of 95.1%, showing its effectiveness in accurately identifying software defects. The framework also evaluates its efficiency by calculating execution times. Notably, it exhibits enhanced efficiency, significantly reducing the execution times during the training and testing phases by an average of 51.52% and 52.31%, respectively. This reduction contributes to a more computationally economical solution for accurate software defect prediction.

Corresponding authors
Tehseen Mazhar,
tehseenmazhar719@gmail.com
Muhammad Amir Khan,
amirkhan@uitm.edu.my

## INTRODUCTION

The world is becoming a global village, and the software industry's primary focus is process improvement and automation. Software applications are the backbone of this global village. A quality software product is defect-free and delivered using minimum resources (*Omri & Sinz, 2020*). The three most important factors to ensure software quality during the development life cycle are time, money, and manpower. The software development life cycle is a multi-stage process. Testing is one of the critical stages that can help ensure that the software is of high quality and free of defects before it is deployed to the production environment (*Liu et al., 2023b*; *Hou et al., 2023*). Software defect prediction is a process that feeds historical defect datasets to the machine learning classifiers that predict which modules are likely to contain defects (*Qiao et al., 2020*); hence, only defective modules should be passed on to the testing stage. The presence of irrelevant features in the dataset can harm the performance of software defect prediction (SDP) models (*Bindu & Sabu, 2020*). Therefore, it is essential to carefully select and pre-process only the meaningful features (*Xiaolong, Wen & Xinheng, 2021*; *Zhou & Zhang, 2022*; *Liu et al., 2023a*; *Zhang et al., 2023*). In addition to choosing the most suitable features, classification techniques have an essential role in SDP because they can help to identify and address potential defects (*Shah & Pujara, 2020*). Previous classification techniques implemented for SDP have been ineffective due to over fitting/under fitting or inadequate results (*Goyal, 2022*; *Liu, Wang & Wang, 2021*). Hence, there is a need to choose a combination of classifiers that outperforms the former classification approaches. Researchers have also experimented with ensemble classification techniques but are vulnerable to biases (*Kaur & Kaur, 2021*). To deal with all these issues, a framework is required that addresses the problem of SDP efficiently, engaging fewer resources and in a cost-effective manner. For classification, random forest (RF), support vector machine (SVM), and naive Bayes (NB) are popular classifiers that are used extensively to tackle the software defect prediction problem. Furthermore, the effectiveness of classification techniques can be boosted by incorporating ensemble techniques. Voting is an effective ensemble technique employed for software defect prediction as it combines the predictions of multiple base classifiers to improve overall accuracy and robustness (*Tewari & Dwivedi, 2020*).

This research proposes a comprehensive five-stage framework for software defect prediction, aiming to address the critical need for efficient and cost-effective solutions. This framework integrates feature selection using a genetic algorithm, implementing heterogeneous binary classifiers, applying ensemble techniques, and evaluating performance using comprehensive measures. The primary contribution is found in the feature selection stage, where accuracy is enhanced, computational costs are reduced, and interpretability is improved compared to existing techniques (*Iqbal et al., 2019a*; *Aljamaan & Alazba, 2020*; *Alsghaier & Akour, 2020*).

This research aims to develop an efficient pre-testing solution for defect prediction using classification techniques by achieving the following objectives:

- To enhance predictive power by optimizing feature selection with genetic search for effective defect prediction

- To improve the accuracy of the software defect prediction system by developing an ensemble-based classification framework
- To compare the performance of the proposed framework with state-of-the-art techniques to showcase its effectiveness

## Motivation of the study

Software quality assurance (SQA) aims to ensure software quality throughout its development life cycle. A critical SQA task is the early identification of defective modules, as addressing defects in later stages is resource-intensive (*Iqbal et al., 2019b*). Effective defect prediction relies on optimal feature selection from historical software data. Previous studies have demonstrated that feature selection techniques can enhance classifier accuracy (*Balogun et al., 2021*; *Singh & Haider, 2022*). However, classification techniques alone have not consistently delivered exceptional results. By integrating diverse classification techniques, Ensemble learning crucially enhances predictive accuracy and robustness (*Jacob et al., 2021*; *Ali et al., 2020*; *Alazba & Aljamaan, 2022*). Recent research (*Iqbal & Aftab, 2020*) in software defect prediction using machine learning classifiers; has revealed varying accuracy rates—ranging from 79.59% for CM1 to 74.8% for PC4—indicating suboptimal outcomes in some instances (62.78% for JM1, 62.16% for MC2, 77.33% for MW1, 89.65% for PC1, and 75.94% for PC3). Therefore, there is a demand for a framework that combines feature selection *via* genetic algorithms with heterogeneous classification methods like RF, SVM, and NB, along with a voting ensemble approach, to achieve higher accuracy.

## Organization of the study

This study has been organized as follows: 'Literature Review' provides an overview of existing research in the field through the literature review. 'Materials and Methods' outlines the proposed framework, with detailed descriptions of its stages. 'Results and Discussion' encompasses an extensive analysis and discussion of the results obtained by applying the proposed framework. 'Threats to Validity' discusses the potential validity concerns associated with the research. Finally, 'Conclusion' concludes the study by concisely summarizing the findings and directions for future research.

# LITERATURE REVIEW

Software defect prediction is crucial in the SDLC, identifying modules requiring detailed testing. ML techniques, mainly supervised and unsupervised, are commonly used for prediction purposes alongside other ML categories like semi-supervised and reinforcement learning (*Long et al., 2023*). Figure 1 shows the categorization of ML techniques. Among these techniques, classification techniques are highly prevalent for software defect prediction. These techniques are often coupled with feature selection to enhance accuracy by selecting optimal features and eliminating those hindering performance.

## Feature selection

Regarding the prediction of software defects, a hybrid framework using various classification algorithms was developed by implementing twelve NASA datasets (*Iqbal et al., 2019b*). The

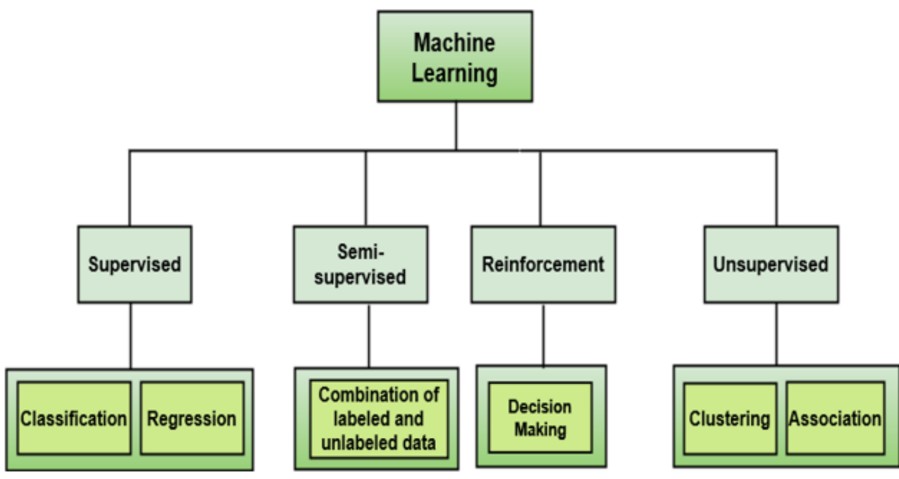

**Figure 1** Machine learning (ML) categories.

experiment had two individual approaches; the first was with feature selection, and the other was without feature selection. Each approach implemented two ensemble strategies, *i.e.,* bagging and boosting, taking RF as the base classifier, improving the framework's accuracy. Similarly, a researcher in *Balogun et al. (2019a)* explored filter feature ranking along with fourteen filter subset selection (FSS) techniques with five NASA datasets and best first search (BFS) as an FS technique. They concluded that the FS strategy generates results with better prediction accuracy and that the feature filter ranking methods were more stable for prediction purposes. Another researcher worked on FS and studied eight FS techniques using five supervised and five unsupervised learning models to cross-check the variance in performance. Both learning models performed better when implemented with the FS technique. They observed that the results of neural network-based techniques are better for unsupervised learning models, while consistency & and correlation-based FS methods work better for supervised learning models (*Kondo et al., 2019*). There are several ML techniques for FS; among them, the Genetic Algorithm (GA) is a robust algorithm used to solve optimization problems (*Ayon, 2019*). This algorithm is closer to nature, is based on the natural process of evolution, and is efficient in solving computationally expensive problems (*Hamdia, Zhuang & Rabczuk, 2021*). A recent study reviewed genetic algorithms and their variants, mainly focusing on multimedia and wireless network applications (*Katoch, Chauhan & Kumar, 2021*). Another FS technique using a Genetic search and encoder–decoder (E–D) model having long short-term memory (LSTM) was implemented to forecast air pollution particulate matter (PM) 2.5 using datasets taken from Hanoi and Taiwan. The E–D model generated results showing improved accuracy (*Nguyen et al., 2021*).

## Classification

Classification, a supervised machine learning technique, is widely applied in various prediction models, including weather forecasting, disease prediction, sentiment analysis, and software defect prediction (*Iqbal et al., 2019b; Luo et al., 2022; Li, Ortegas & White,*

*2023*). In software defect prediction (SDP), a comparative analysis of four classifiers was conducted using NASA's datasets (*Daoud et al., 2022*). The fused artificial neural network Bayesian regularization (ANN-BR) classifier, particularly the Bayesian regularization (BR) classifier, demonstrated remarkable performance. *Iqbal et al. (2019a)* expanded this research by analyzing twelve datasets from NASA, employing multiple classifiers, and comparing results using various performance measures. Across different datasets and algorithms, significant variations in classification performance were observed. For example, the JM1 dataset excelled when paired with the RBF algorithm, while KC3 demonstrated improved results with the MLP algorithm. MC1 achieved remarkable results using KStar, and PC2 also performed well with KStar. In another study, ten classifiers were evaluated for SDP, with the random forest classifier showing enhanced performance on the PROMISE datasets (*Cetiner & Sahingoz, 2020*). Additionally, a software defect prediction model utilizing LASSO-SVM with a reduced-dimension dataset was proposed, incorporating cross-validation to enhance model performance (*Wang et al., 2021*). In *Iqbal (2019)*, researchers developed a model using artificial neural networks and conducted an empirical comparison of backpropagation training algorithms, including Liebenberg-Marquardt, Bayesian Regularization, Scaled Conjugate Gradient, and BFGS Quasi-Newton techniques, on CM1, KC3, MW1, PC1, and PC2 datasets from NASA and remarkable accuracies were achieved on all the employed datasets.

## Ensemble learning

Ensemble learning (EL) is a technique in ML in which predictions from several weak classifiers are integrated to produce a strong classifier that generates better results than standalone classifiers (*Mehta & Patnaik, 2021*). EL offers a range of homogeneous classifiers like bagging, boosting, rotational forest, *etc.*, and several heterogeneous classifiers, including voting, stacking, *etc.* (*Wu & Wang, 2023*). Voting is a heterogeneous ensemble classifier that combines predictions from diverse base classifiers. A software defect prediction framework was proposed by a researcher in *Javed (2021)* using a nested EL technique, *i.e.*, voting as the master classifier, and three base ensemble classifiers, *i.e.*, bagging, boosting, and stacking. The accuracy produced by the proposed framework on two different NASA datasets was 83.46% and 79.65%. Moreover, a software defect prediction model was developed by a researcher in *Matloob (2020)* using multi-layer feed-forward neural networks and stacking as an ensemble technique. Six search methods were implemented for feature selection and multilayer perceptron was used as a subset evaluator. The accuracy achieved on NASA's datasets using best-first search, greedy stepwise search, and genetic search was 80%, 75%, and 76% respectively.

## Limitations of previous research

The existing studies in software defect prediction have made noteworthy contributions; however, several limitations have been identified, particularly low accuracy. Authors in *Amin (2019)* implemented iterative feature selection using SMOTE and BORUTA approaches with SVM and neural networks as classifiers, achieving a maximum accuracy of 76% on NASA's datasets. However, the absence of ensemble learning techniques in

the methodology is notable, potentially introducing bias and low accuracy, limiting the overall predictive capabilities. *Iqbal et al. (2019a)* analyzed twelve datasets from NASA, employing various classifiers such as naïve Bayes, multilayer perceptron (MLP), radial basis function (RBF), SVM, K nearest neighbor (KNN), kStar, OneR, PART, decision tree (DT), and random forest (RF). While showcasing diverse classifiers, the study lacked the incorporation of feature selection and ensemble learning methods. This absence may hinder the model's generalization of datasets, impacting predictive accuracy. Similarly, the authors in *Iqbal (2019)* developed a model using artificial neural networks, comparing backpropagation algorithms empirically. Despite achieving notable accuracies ranging from 81.70% to 86.85% on different NASA datasets, the study lacked the incorporation of feature selection and ensemble learning methods with a combination of diverse classifiers. This limitation may affect the model's robustness and generalization to diverse software defect scenarios.

To address these limitations, a framework is proposed for software defect prediction that implements a genetic algorithm to perform feature selection and voting as an ensemble learning technique with a heterogeneous combination of RF, SVM, and NB as base classifiers for enhancing the prediction power of software defect prediction.

The summary of the literature review is presented in Table 1. It outlines the techniques proposed for SDP, the source of datasets used for experimentation, the specific datasets employed, and the performance measures employed to analyze the results.

## MATERIALS AND METHODS

This research introduces an intelligent ensemble-based software defect prediction framework that significantly enhances predictive accuracy by integrating a genetic algorithm-based feature selection technique. The proposed framework uses the collective strengths of diverse supervised machine learning classifiers. The IECGA framework comprises two layers: training and testing. The training layer comprises three stages: (1) feature selection, (2) base classification, and (3) ensemble classification. In the training layer, the process unfolds through three sequential stages: feature selection, conducted with a genetic algorithm, followed by base classification employing random forest, support vector machine, and naive Bayes. Finally, the predictive accuracy of base classifiers is skillfully combined through the voting ensemble classifier, contributing to developing the IECGA framework. The testing layer encompasses a single stage, specifically prediction, which involves predicting defects in new modules using the trained model. A cleaned version of seven widely used, publicly available NASA datasets with a binary prediction class has been applied to implement this framework. Five performance measures have been used, *i.e.,* precision, recall, accuracy, F-measure, and area under the curve (AUC).

The proposed framework is comprised of the following five stages:

Stage 1: Dataset selection

Stage 2: Dataset pre-processing—feature selection

Stage 3: Classification

Stage 4: Ensemble learning

Stage 5: Performance evaluation

**Table 1  Comprehensive summary of literature review.**

| | | | | |
|---|---|---|---|---|
| Goyal (2022) | Feature selection-based classification for SDP using bagging and boosting ensemble techniques | NASA | CM1, MC1,JM1, KC1, KC3, MC2, MW1, PC1, PC2,PC3, PC4, PC5 | Precision, Recall, F-measure, Accuracy, AUC, MCC |
| Liu, Wang & Wang (2021) | Feature selection-based classification for SDP using four algorithms, including NB, DT, LR, and KNN | NASA | CM1, KC1, KC3, MW1, PC2 | Accuracy, Co-efficient of Variation (CV) |
| Kaur & Kaur (2021) | A comparative analysis of two feature selection techniques for SDP: 1) correlation and consistency-based feature selection. 2) Neural network-based feature reduction | PROMISE, NASA, AEEEM | PROMISE(Ant v1.7, Camel v1.6, Ivy v1.4, Jedit v4.0, Log4j v1.0, Lucene v2.4, POI v3.0, Tomcat v6.0, Xalan v2.6, Xerces v1.3) NASA(CM1, JM1, KC3, MC1, MC2, MW1, PC1, PC2,PC3, PC4, PC5) AEEEM(Eclipse JDT Core, Equinox, Apache Lucene, Mylyn, Eclipse PDE UI) | AUC , interquartile range (IQR) |
| Aljamaan & Alazba (2020) | A review of genetic algorithms for feature selection mainly focused on multimedia and wireless network applications. | – | – | – |
| Alsghaier & Akour (2020) | Genetic algorithm-based feature selection model to predict pollution particulate matter (PM) 2.5 | - | Hanoi and Taiwan | – |
| Balogun et al. (2021) | A comparative analysis among four classifiers based on a back propagation strategy for software defect prediction | NASA | CM1, MC1,JM1, KC1, KC3, MC2, MW1, PC1, PC2,PC3, PC4, PC5 | Specificity, precision, Recall, F-measure, AUC, accuracy, R2, mean-square error |
| Singh & Haider (2022) | Performance analysis on ten machine learning classifiers, including NB, MLP, RBF, SVM, KNN, kStar, One Rule (OneR), PART, Decision Tree (DT), and RF | NASA | CM1, MC1,JM1, KC1, KC3, MC2, MW1, PC1, PC2,PC3, PC4, PC5 | Precision, Recall, F-measure, Accuracy, MCC |
| Jacob et al. (2021) | A comparative analysis of machine learning-based software defect prediction systems by analyzing ten supervised classification algorithms | PROMISE | CM1, KC1, KC2, JM1, and PC1 | Accuracy, Precision, Recall |
| Ali et al. (2020) | LASSO –SVM-based model for software defect prediction using a reduced-dimension dataset | NASA | PC1 | Precision, Recall, F-measure, accuracy |
| Iqbal & Aftab (2020) | Nested ensemble learning using voting as a master classifier along with bagging, boosting, and stacking as base classifiers for SDP | NASA | CM1, KC1, KC3, MC1, MW1, PC1, PC2, PC3, PC4, PC5 | F-measure, MCC, Accuracy |
| Long et al. (2023) | multi-layer feed-forward neural networks using stacking as an ensemble technique for SDP | NASA | KC1, KC3, MC2, MW1, PC4 and PC5 | Accuracy, Precision, Recall, F-measure, MCC, AUC |

## Dataset selection

This research utilizes seven NASA datasets, namely CM1, JM1, MC2, MW1, PC1, PC3, and PC4 which are publicly accessible. These datasets were collected from NASA's real software projects and developed in different languages. In Shepperd et al. (2013), researchers

**Table 2  Cleaning criteria.**

| Criterion | Data quality category | Explanation |
|---|---|---|
| 1 | Identical cases | Instances that have identical values for all metrics, including class label |
| 2 | Inconsistent cases | Instances that satisfy all conditions of Case 1 but where class labels differ |
| 3 | Cases with missing values | Instances that contain one or more missing observations |
| 4 | Cases with conflicting feature values | Instances with 2 or more metric values violate some referential integrity constraint. For example, LOC TOTAL is less than Commented LOC. However, Commented LOC is a subset of LOC Total. |
| 5 | Cases with implausible values | Instances that violate some integrity constraint. For example, the value of LOC=1.1 |

**Table 3  Description of NASA's cleaned D″ software defect datasets.**

| Dataset | No. of attributes | No. of modules | Defective modules | Non-defective modules | Defective instances (%) |
|---|---|---|---|---|---|
| CM1 | 38 | 327 | 42 | 285 | 12.8 |
| JM1 | 22 | 7,720 | 1,612 | 6,108 | 20.8 |
| MC2 | 40 | 124 | 44 | 80 | 35.4 |
| MW1 | 38 | 250 | 25 | 225 | 10 |
| PC1 | 38 | 679 | 55 | 624 | 8.1 |
| PC3 | 38 | 1,053 | 130 | 923 | 12.3 |
| PC4 | 38 | 1,270 | 176 | 1,094 | 13.8 |

analyzed NASA's defect datasets and produced two individually cleaned versions of the datasets, *i.e.,* DS' and DS″, the benchmark datasets. The DS dataset contains identical and conflicting values. Meanwhile, D excludes duplicate and inconsistent data. This research uses seven datasets from the DS″ version to implement the proposed framework. Table 2 presents the cleaning criteria used for NASA datasets implemented by *Shepperd et al. (2013)*.

Each dataset consists of many independent attributes, *i.e.,* BRANCH_COUNT, CYCLOMATIC_COMPLEXITY, HALSTEAD_EFFORT, LOC_TOTAL, LOC_EXECUTABLE, *etc.,* and one dependent attribute called the target class. The dependent attribute is predicted based on the independent attributes. The target class contains a Boolean value ''Y'' or ''N'' where ''Y'' represents that the particular module is defective and ''N'' represents that it is non–defective. The dataset was partitioned into training and testing sets following a 70–30 proportion, employing a class-based splitting rule to ensure a representative distribution across classes. The details of the datasets used in this research are given in Table 3.

The target class distribution of the employed datasets is presented in Fig. 2.

## Feature selection

The software consists of many features, but only a few features positively impact the performance of classifiers (*Kondo et al., 2019*; *Ali et al., 2023*). In this research, a filter-based feature selection technique has been used that evaluates all the attributes available in the underlying dataset and draws only those attributes that are more appropriate based on the

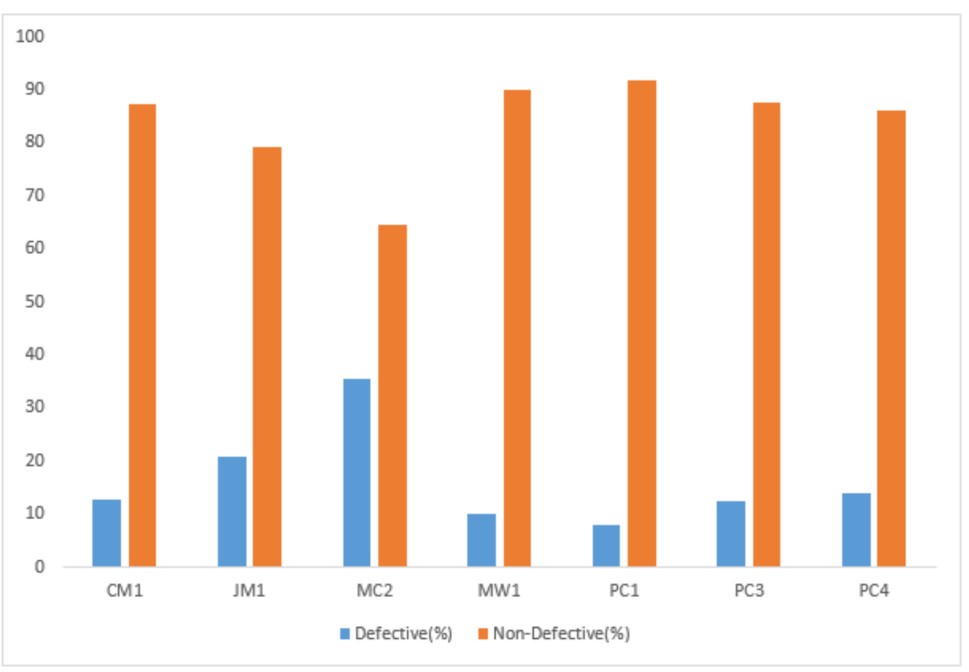

**Figure 2 Class distribution.**

target class. The feature selection technique comprised two parts, *i.e.,* (1) the search method and (2) the attribute evaluation method. This research employs a GA search method and CfsSubsetEval as an attribute evaluation method. GA involves stages like population initialization, fitness function calculation, selection of fittest individuals, crossover, mutation, and termination upon reaching a defined threshold. GA was incorporated with the following parameter values: a population size of 20 individuals, a maximum of 20 generations, a crossover probability set to 0.6, and a mutation probability of 0.033. These parameters were precisely chosen through empirical testing and iterative tuning to optimize the effectiveness and efficiency of the feature selection process. GA's evolutionary approach allows it to efficiently explore diverse feature subsets, aligning well with the feature selection problem. Its capability to handle non-linear relationships within the dataset is particularly advantageous, contributing to identifying informative features while mitigating computational costs (*Bindu & Sabu, 2020*; *Maleki, Zeinali & Niaki, 2021*). The adaptability of GA to problem complexity adds a robust dimension to the feature selection process, enhancing the overall predictive power of the framework (*Hamdia, Zhuang & Rabczuk, 2021*). Hence, the algorithm identifies and returns the best-performing individuals crucial for problem-solving (*Peng et al., 2021*). This research employs the Correlation-based Feature Selection (CFS) method, a filter technique for evaluating each attribute's predictive power concerning the output class. Attributes with the highest correlation to the output class are selected while minimizing inter-correlation among dataset attributes (*Zhu et al., 2021*). GA-driven feature selection identified an optimal subset of features, ensuring that the most informative attributes were retained through an iterative selection process, crossover,

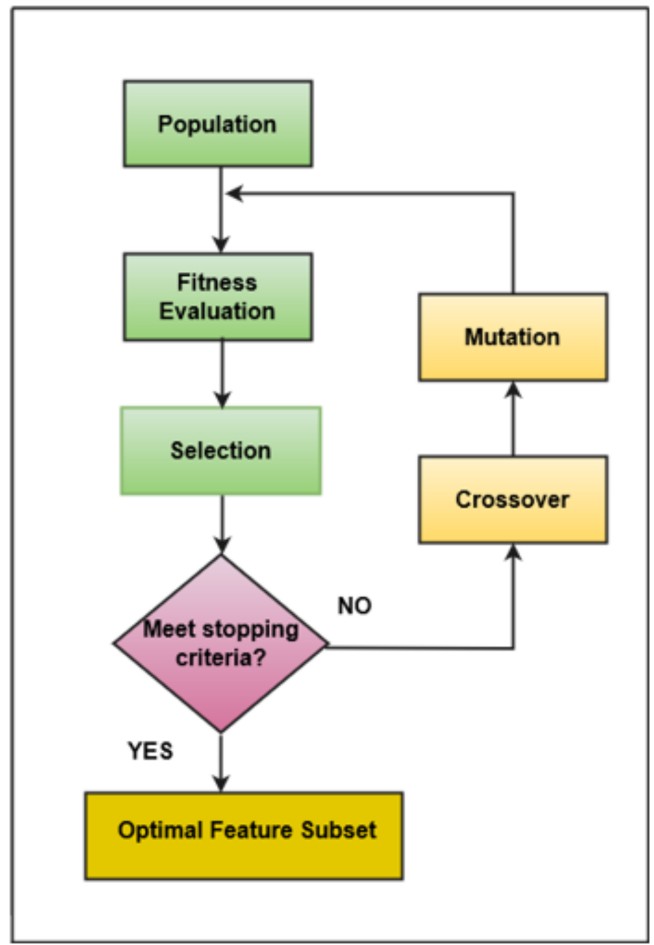

**Figure 3  Genetic algorithm.**

**Table 4  Feature subset generated after feature selection.**

| Datasets | CM1 | JM1 | MC2 | MW1 | PC1 | PC3 | PC4 |
|---|---|---|---|---|---|---|---|
| Original features | 38 | 22 | 40 | 38 | 38 | 38 | 38 |
| Selected features | 8 | 10 | 10 | 8 | 7 | 9 | 9 |

and mutation. This process dynamically explored the solution space to converge upon subsets that maximize the classifier's predictive accuracy. The graphical representation of selecting the most optimal features using a genetic algorithm is shown in Fig. 3.

The features produced after applying the genetic algorithm are listed in Table 4.

A thorough feature correlation analysis was conducted for subsequent analysis to gain deeper insights into the dataset and understand the relationships among different features. This analysis is crucial in revealing potential dependencies or redundancies among the selected features. The feature selection process, which leveraged a genetic algorithm search method in conjunction with the CfsSubsetEval evaluator, aimed to identify the

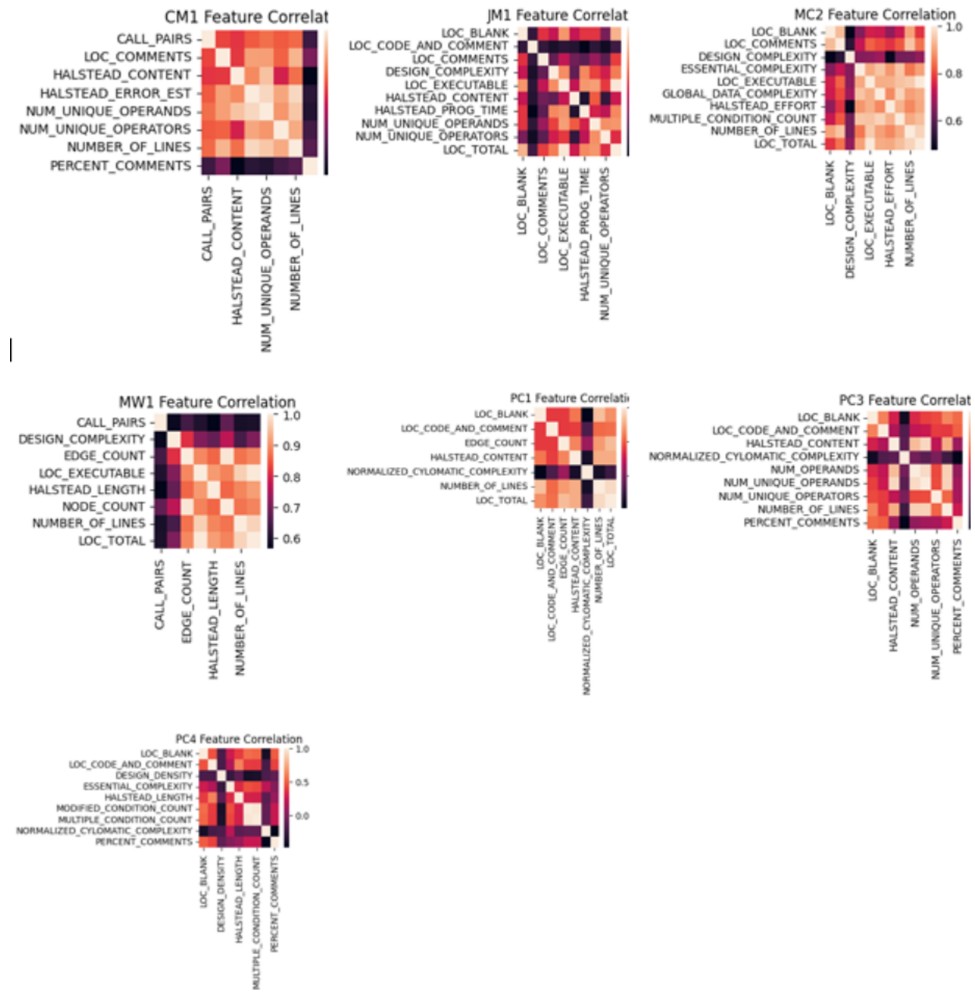

**Figure 4   Correlation graph for NASA datasets.**

optimal subset of features for the given task. Before showcasing the results of this feature selection process, examining the degree of correlation between the features themselves is imperative. Correlation diagrams visually represent these relationships and play a pivotal role in the decision-making process when selecting our predictive model's final set of features. By unveiling feature correlations, this analysis ensures that the selected features are both informative and independent, thus contributing to the overall effectiveness of our software defect prediction framework. The correlation graph for each training dataset is presented in Fig. 4, which uses color intensity to represent the strength and direction of feature correlations. Darker shades indicate stronger correlations, with positive correlations represented by warmer and negative correlations by cooler colors.

## Classification

This research implies three classification algorithms of heterogeneous nature, *i.e.,* RF, SVM, and NB, that produced significant results in the development of a software defect prediction framework. The description of each classifier is given as follows:

**Random forest (RF):** RF utilizes the strength of decision trees, where each tree is trained on a subset of the dataset and makes independent predictions. The final output is determined through a voting mechanism, providing a more robust and accurate prediction (*Soe, Santosa & Hartanto, 2018*; *Ibrahim, Ghnemat & Hudaib, 2017*). This approach enhances predictive accuracy and fosters resilience against overfitting and variability in software defect patterns (*Alshammari, 2022*; *Mafarja et al., 2023*). In mathematical terms, we can describe RF as an ensemble of decision trees:

$$RF(X) = Tree\_1(X) + Tree\_2(X) + \cdots + Tree\_N(X). \tag{1}$$

where RF(X) represents the Random Forest's prediction for input data point X. Tree_1(X), Tree_2(X), …, Tree_N(X) represent the predictions of individual decision trees in the forest. Each tree _i (X) is the output of an individual decision tree, which can be a complex, recursive structure involving feature selection, node splitting, and class prediction.

**Support vector machine (SVM):** SVM operates by identifying a hyperplane that best separates different classes in the feature space, maximizing the margin between them (*Wang et al., 2021*; *Kumar & Singh, 2017*). By leveraging the principles of margin maximization, SVM enhances the framework's ability to distinguish complicated patterns, providing a valuable addition to accurate software defect prediction (*Mustaqeem & Saqib, 2021*). SVM classifier can be represented in mathematical form as follows:

Given a dataset with feature vectors x_i ($i = 1, 2, …, N$) and corresponding binary labels y_i (y_i $\in$ {-1, 1}), the goal of SVM is to find a decision boundary in the feature space that maximizes the margin between the two classes while minimizing classification errors (*Husin, Pribadi & Yohannes, 2022*).

The decision boundary can be defined as:

$$f(x) = sign(\sum [alpha\_i.y\_i(x\_i.x) + b]). \tag{2}$$

where $f(x)$ is the decision boundary function for classifying a new data point $x$, alpha_i represents the Lagrange multipliers associated with each data point, $x\_i$ represents the support vectors (data points that lie on the margin or are misclassified), $x \cdot x\_i$ represents the dot product between the feature vectors $x$ and $x\_i$, $b$ is a constant bias term.

In the case of a polynomial kernel of degree 2 (quadratic), the kernel function $K(x, x\_i)$ is defined as:

$$k(x, x_i) = (x.x_i + 1)2. \tag{3}$$

So, the decision boundary for the SVM with a polynomial kernel of degree 2 ($C = 2$) would involve using this kernel function within the decision boundary equation:

$$f(x) = sign\left(\sum [alpha\_i.y\_i.(k(x, x\_i)) + b]\right). \tag{4}$$

**Naive Bayes (NB):** NB is a probabilistic classification algorithm that works on the principles of Bayes' theorem. It is based on the assumption of feature independence, meaning that each feature in the dataset is considered unrelated to others, given the class label (*Rahim et al., 2021*; *Hernández-Molinos et al., 2023*). Naive Bayes efficiently categorizes instances by calculating the likelihood of each class given the observed features (*Tua & Danar Sunindyo, 2019*). Its simplicity, efficiency, and ability to handle diverse feature sets make it valuable in our framework. The mathematical form of Gaussian Naive Bayes can be simplified as follows:

$$P(y|x) = (1/(sqrt(2*\pi*\sigma^2)))*e^(-((x-\mu)^2/(2*\sigma^2))). \tag{5}$$

where $P(y|x)$ represents the probability of class $y$ given the feature vector $x$, $\mu$ is the mean (average) of the feature values for class $y$, $\sigma^2$ is the variance (spread) of the feature values for classy, e is the base of the natural logarithm, approximately equal to 2.71828.

Machine learning classifiers consist of parameters that can be adjusted to find optimized values. A systematic tuning approach was adopted to determine the optimal hyperparameters for each classifier. Initially, a hit-and-trial method was employed to explore a broad range of hyperparameter values for each classifier. Subsequently, an iterative tuning process was conducted, refining the hyperparameter values based on the performance metrics observed during multiple trial runs. RF showed significant results when the *max depth* parameter was set to "10", and the *n_estimators* parameter was set to "500". SVM showed the optimal performance when the *kernel* parameter was chosen as "poly", and the Regularization Parameter $C$ was set to "2". NB achieved the highest performance when "GaussianNB" was implemented. The process of classifier tuning is shown in Fig. 5. The combination of RF, ', and NB was deliberately chosen due to their diverse and heterogeneous nature and complementary strengths. Each classifier contributes its unique characteristics: RF is based on decision trees; SVM employs a probability-based approach; and NB is rooted in Bayes' theorem (*Ibrahim, Ghnemat & Hudaib, 2017*; *Tua & Danar Sunindyo, 2019*; *Azzeh et al., 2023*).

## ENSEMBLE LEARNING

Ensemble classification combines multiple base classifiers to enhance accuracy and stability. This research employs a voting algorithm as a heterogeneous ensemble classifier to improve framework accuracy. Voting ensembles can be categorized as hard voting, where the output class label is determined by majority votes, and soft voting, where the prediction probability of each classifier informs the final prediction (*Ali et al., 2020*). Soft voting is implemented in this research to boost software defect prediction performance. Figure 6 presents the IECGA framework for software defect prediction, providing a visual overview of the critical components and stages of our proposed methodology.

The source code file and the datasets employed in the IECGA framework have been publicly posted on GitHub (https://github.com/misbah-here/IECGA-Framework).

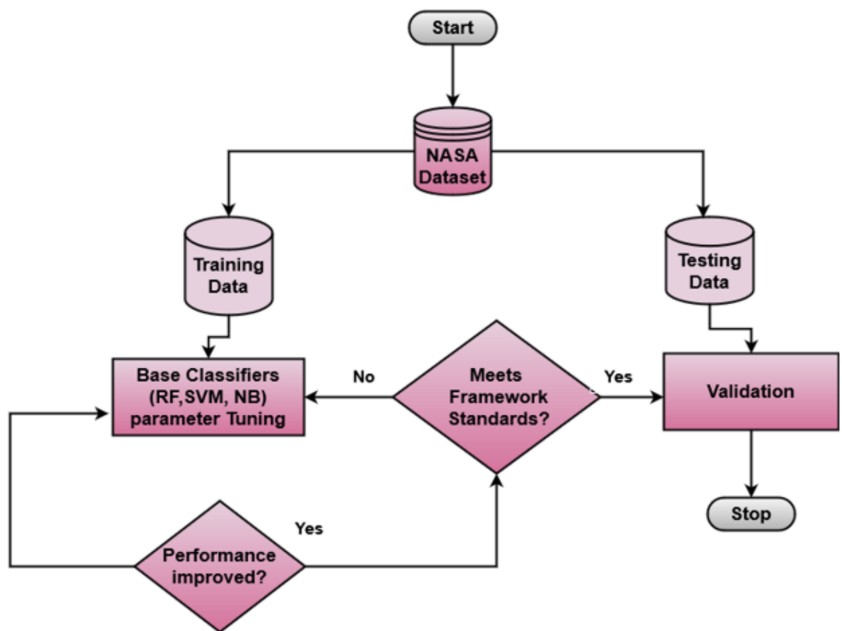

**Figure 5  Classifiers tuning.**

## Performance evaluation

After implementing the proposed framework, the most critical task is to examine its performance. Various machine learning tools provide several evaluation measures that comprehensively present performance, such as recall, precision, and G-mean (*Iqbal, 2019*). The best approach to evaluate the framework's efficiency is to organize the results using a confusion matrix (*Alsawalqah et al., 2020*). To evaluate the effectiveness of the proposed approach, five widely recognized evaluation measures, namely precision, recall, accuracy, *F*-measure, and AUC, were thoughtfully applied (*Balogun et al., 2019a*).

Precision measures the accuracy of optimistic predictions, expressing the ratio of true positives to the total predicted positives. It is particularly relevant when minimizing false positives, which is crucial. Recall evaluates the model's ability to capture all relevant instances. It is calculated as the ratio of true positives to actual positives. Accuracy is a fundamental metric representing the overall correctness of the model. It is calculated as the ratio of correctly predicted instances to the total instances, providing a holistic view of the model's performance (*Balogun et al., 2019a*). The F-measure is the harmonic mean of precision and Recall. It offers a balanced assessment by considering both false positives and false negatives. AUC assesses the model's ability to distinguish between positive and negative instances across various threshold levels. A higher AUC indicates superior discrimination capability, commonly used in binary and multiclass classification tasks (*Lear et al., 2021*). These measures were derived using a confusion matrix and were calculated with Python functions. The formula used to calculate each performance measure is given as:

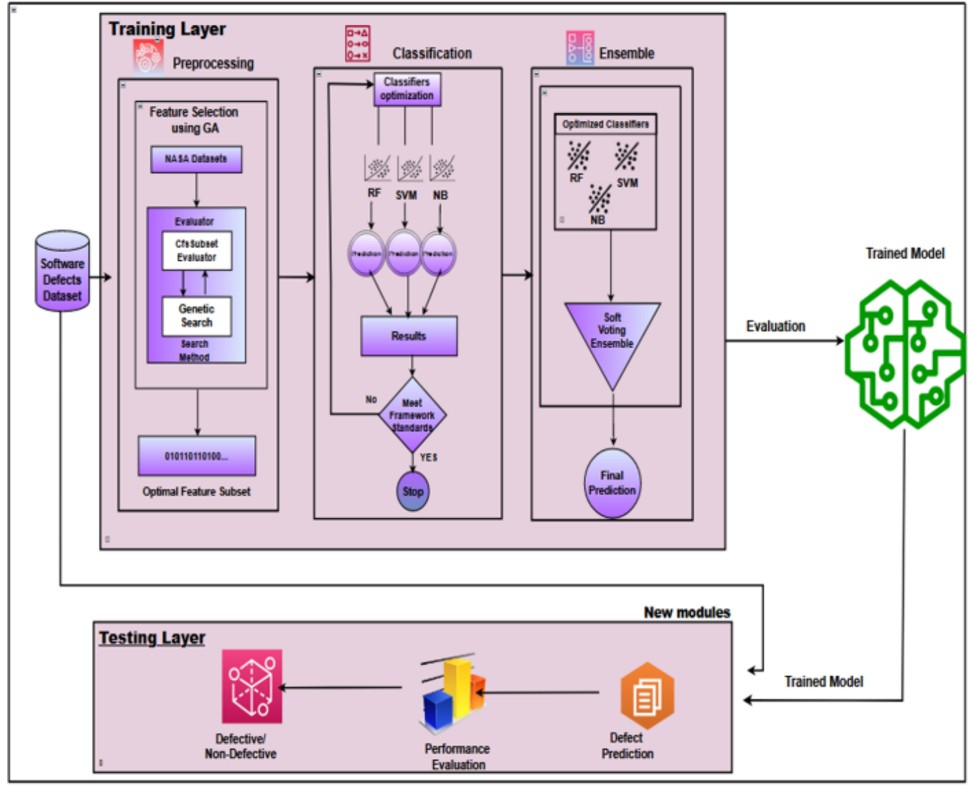

**Figure 6** IECGA framework for software defect prediction.

$$Precision = \frac{TP}{TP + FP} \tag{6}$$

$$Recall = \frac{TP}{TP + FN} \tag{7}$$

$$Accuracy = \frac{(TP + TN)}{(TP + TN + FP + FN)} \tag{8}$$

$$F - measure = \frac{2 * Recall * Precision}{Recall + Precision} \tag{9}$$

$$AUC = \frac{1 + TPr - FPr}{2}. \tag{10}$$

In the provided equations, TP corresponds to the accurate identification of defective software modules by the model, while TN signifies the correct recognition of non-defective software modules. The values FP and FN indicate discrepancies between the actual and predicted outcomes. Specifically, FP signifies instances where a non-defective module was predicted as defective. In contrast, FN indicates cases where a module, actually defective, was predicted as non-defective by the model. In contrast, TPr represents the actual positive rate, measuring the proportion of correctly predicted positive instances among all actual positives. FPr represents the false positive rate, indicating the proportion of incorrectly predicted negative instances among all negatives.

**Table 5   Detailed results of classifiers on the CM1 dataset.**

| ML classifier | Dataset | Precision | Recall | Accuracy (%) | F-Measure | AUC |
|---|---|---|---|---|---|---|
| RF | Training | 1 | 1 | 100 | 1 | 1 |
| | Testing | 0.25 | 0.08 | 84.85 | 0.12 | 0.63 |
| SVM | Training | 1 | 0.1 | 88.6 | 0.19 | 0.35 |
| | Testing | 1 | 0.08 | 87.88 | 0.14 | 0.42 |
| NB | Training | 0.43 | 0.41 | 85.53 | 0.42 | 0.74 |
| | Testing | 0.38 | 0.23 | 84.85 | 0.29 | 0.73 |
| IECGA | Training | 1 | 0.38 | 92.11 | 0.55 | 0.96 |
| | Testing | 0.33 | 0.08 | 85.86 | 0.12 | 0.65 |

**Table 6   Detailed results of classifiers on the JM1 dataset.**

| ML Classifier | Dataset | Precision | Recall | Accuracy% | F-Measure | AUC |
|---|---|---|---|---|---|---|
| RF | Training | 1 | 0.37 | 86.77 | 0.54 | 0.93 |
| | Testing | 0.61 | 0.15 | 80.18 | 0.24 | 0.71 |
| SVM | Training | 0.86 | 0.01 | 79.22 | 0.01 | 0.67 |
| | Testing | 1 | 0.01 | 79.49 | 0.03 | 0.67 |
| NB | Training | 0.49 | 0.21 | 79 | 0.29 | 0.64 |
| | Testing | 0.52 | 0.2 | 79.49 | 0.29 | 0.63 |
| IECGA | Training | 0.74 | 0.2 | 81.83 | 0.32 | 0 |
| | Testing | 0.55 | 0.16 | 79.66 | 0.25 | 0.71 |

# RESULTS AND DISCUSSION

In this research, an intelligent ensemble-based software defect prediction framework named IECGA was implemented. To perform the experiments, seven publicly accessible NASA datasets (CM1, JM1, MC2, MW1, PC1, PC3, and PC4) were extracted from the MDP repository. In the pre-processing step, feature selection was performed using genetic algorithms, enriching the model's predictive capabilities. Following the class-based splitting rule, the datasets were subsequently divided into training and testing subsets using 70:30 ratios (*Kaur & Kaur, 2021*). Initially, three heterogeneous supervised classification algorithms were employed to train the model: RF, SVM, and NB. These classifiers underwent iterative optimization to maximize their accuracy for the selected datasets. The predictive accuracy from individual classifiers was carefully integrated using a voting ensemble technique, further boosting the model's performance. To evaluate the effectiveness of the proposed approach, five widely recognized evaluation measures, namely precision, Recall, accuracy, F-measure, and AUC, were thoughtfully applied (*Iqbal et al., 2019a*; *Mumtaz et al., 2021*). These measures were derived using a confusion matrix and were calculated with Python functions. The results obtained from training and testing datasets for each dataset are systematically presented in Tables 5–11, emphasizing the impact of feature selection using genetic algorithms on the enhancement of software defect prediction.

Results achieved from the CM1 dataset are presented in Table 5. It reveals that RF excelled in precision and recall during training, while SVM showed high precision but

**Table 7  Detailed results of classifiers on the MC2 dataset.**

| ML Classifier | Dataset | Precision | Recall | Accuracy (%) | F-Measure | AUC |
|---|---|---|---|---|---|---|
| RF | Training | 1 | 1 | 100 | 1 | 1 |
| | Testing | 0.55 | 0.46 | 68.42 | 0.5 | 0.62 |
| SVM | Training | 0.8 | 0.13 | 67.44 | 0.22 | 0.74 |
| | Testing | 1 | 0.15 | 71.05 | 0.27 | 0.61 |
| NB | Training | 0.77 | 0.32 | 72.09 | 0.45 | 0.7 |
| | Testing | 0.57 | 0.31 | 68.42 | 0.4 | 0.58 |
| IECGA | Training | 0.83 | 0.32 | 73.26 | 0.47 | 0.96 |
| | Testing | 0.67 | 0.31 | 71.05 | 0.42 | 0.62 |

**Table 8  Detailed results of classifiers on the MW1 dataset.**

| ML Classifier | Dataset | Precision | Recall | Accuracy % | F-Measure | AUC |
|---|---|---|---|---|---|---|
| RF | Training | 1 | 1 | 100 | 1 | 1 |
| | Testing | 0.5 | 0.25 | 89.33 | 0.33 | 0.8 |
| SVM | Training | 1 | 0.12 | 91.43 | 0.21 | 0.71 |
| | Testing | 0 | 0 | 89.33 | 0 | 0.95 |
| NB | Training | 0.32 | 0.53 | 84.57 | 0.4 | 0.71 |
| | Testing | 0.4 | 0.75 | 85.33 | 0.52 | 0.88 |
| IECGA | Training | 1 | 0.47 | 94.86 | 0.64 | 0.95 |
| | Testing | 0.5 | 0.25 | 89.33 | 0.33 | 0.86 |

**Table 9  Detailed results of classifiers on the PC1 dataset.**

| ML Classifier | Dataset | Precision | Recall | Accuracy % | F-Measure | AUC |
|---|---|---|---|---|---|---|
| RF | Training | 1 | 1 | 100 | 1 | 1 |
| | Testing | 0.86 | 0.35 | 94.12 | 0.5 | 0.91 |
| SVM | Training | 1 | 0.16 | 93.26 | 0.27 | 0.79 |
| | Testing | 0.67 | 0.24 | 92.65 | 0.35 | 0.63 |
| NB | Training | 0.23 | 0.34 | 85.68 | 0.28 | 0.82 |
| | Testing | 0.48 | 0.71 | 91.18 | 0.57 | 0.91 |
| Proposed IECGA | Training | 1 | 0.34 | 94.74 | 0.51 | 0.94 |
| | Testing | 0.89 | 0.47 | 95.1 | 0.62 | 0.93 |

lower recall. NB achieved good overall accuracy with balanced precision and recall. The IECGA framework exhibited impressive precision, decent recall, and excellent F-measure during training, proving its effectiveness. Although Recall decreased slightly in testing, it maintained solid precision, a reasonable F-Measure, and competitive accuracy. This underscores the framework's balanced and effective software defect prediction, highlighting its potential for enhancing software quality assurance.

The graphical representation of all performance measures on the CM1 testing dataset is shown in Fig. 7.

**Table 10  Detailed results of classifiers on the PC3 dataset.**

| ML Classifier | Dataset | Precision | Recall | Accuracy % | F-Measure | AUC |
|---|---|---|---|---|---|---|
| RF | Training | 1 | 1 | 100 | 1 | 1 |
|  | Testing | 0.67 | 0.21 | 88.92 | 0.31 | 0.8 |
| SVM | Training | 1 | 0.03 | 88.06 | 0.06 | 0.7 |
|  | Testing | 0.25 | 0.03 | 87.03 | 0.05 | 0.68 |
| NB | Training | 0.34 | 0.58 | 80.73 | 0.43 | 0.82 |
|  | Testing | 0.33 | 0.54 | 81.01 | 0.41 | 0.77 |
| IECGA | Training | 1 | 0.51 | 93.89 | 0.67 | 0.95 |
|  | Testing | 0.57 | 0.21 | 88.29 | 0.3 | 0.81 |

**Table 11  Detailed results of classifiers on the PC4 dataset.**

| ML Classifier | Dataset | Precision | Recall | Accuracy % | F-Measure | AUC |
|---|---|---|---|---|---|---|
| RF | Training | 1 | 1 | 100 | 1 | 1 |
|  | Testing | 0.62 | 0.43 | 88.45 | 0.51 | 0.91 |
| SVM | Training | 1 | 0.03 | 86.61 | 0.06 | 0.78 |
|  | Testing | 1 | 0.02 | 86.35 | 0.04 | 0.7 |
| NB | Training | 0.53 | 0.39 | 86.73 | 0.45 | 0.83 |
|  | Testing | 0.65 | 0.87 | 88.71 | 0.51 | 0.87 |
| IECGA | Training | 0.96 | 0.38 | 91.23 | 0.55 | 0.98 |
|  | Testing | 0.82 | 0.34 | 89.76 | 0.48 | 0.91 |

Results achieved from the JM1 dataset are presented in Table 6. It indicates that RF had perfect precision during training but lower Recall. SVM had high precision but very low Recall, prioritizing false positive reduction. NB had moderate precision and Recall. IECGA had competitive precision and Recall during training. In testing, RF maintained a good balance, SVM prioritized precision, NB remained consistent, and IECGA demonstrated potential, albeit with slightly reduced Recall.

The graphical representation of all performance measures on the JM1 testing dataset is shown in Fig. 8.

Results obtained from the MC2 dataset are shown in Table 7. It reveals that RF achieved perfect precision and recall during training, resulting in flawless accuracy and F-measure. SVM showed good precision but lower recall, ensuring reasonably high accuracy. NB maintained balanced precision and Recall, leading to decent overall accuracy and F-measure. IECGA demonstrated competitive precision and recall in training. In testing, RF maintained good precision and recall, while SVM prioritized precision over recall. NB maintained its balance, and IECGA remained competitive, although with a slight decrease in recall compared to training.

The graphical representation of all performance measures on the MC2 testing dataset is shown in Fig. 9.

Results obtained from the MW1 dataset are displayed in Table 8. It shows that in the training phase, RF achieved perfect precision and recall, indicating its strong ability to identify defects. SVM had perfect precision but lower recall, while NB achieved balanced

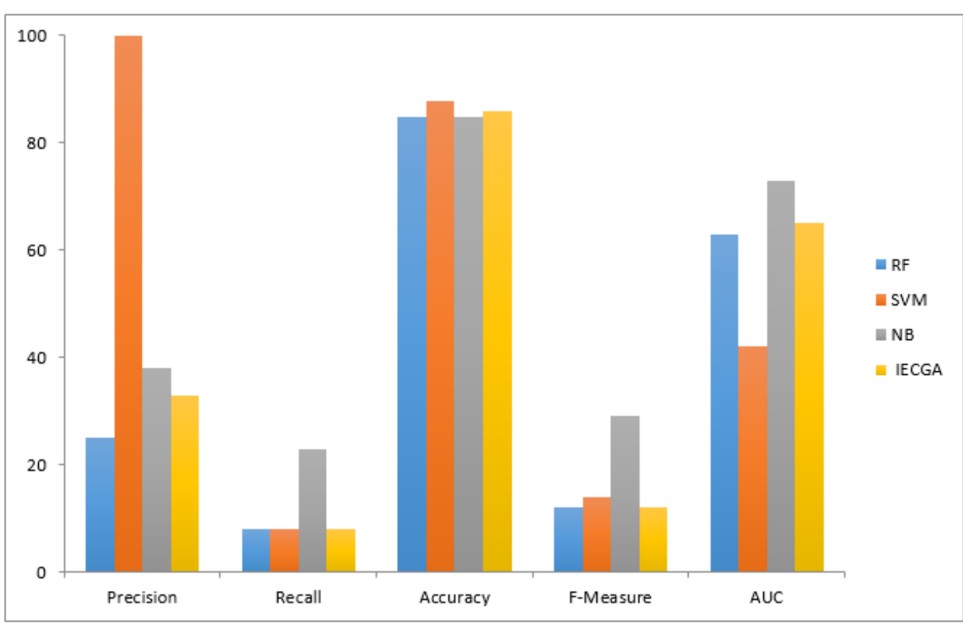

**Figure 7** **Performance measures on the CM1 testing dataset.**

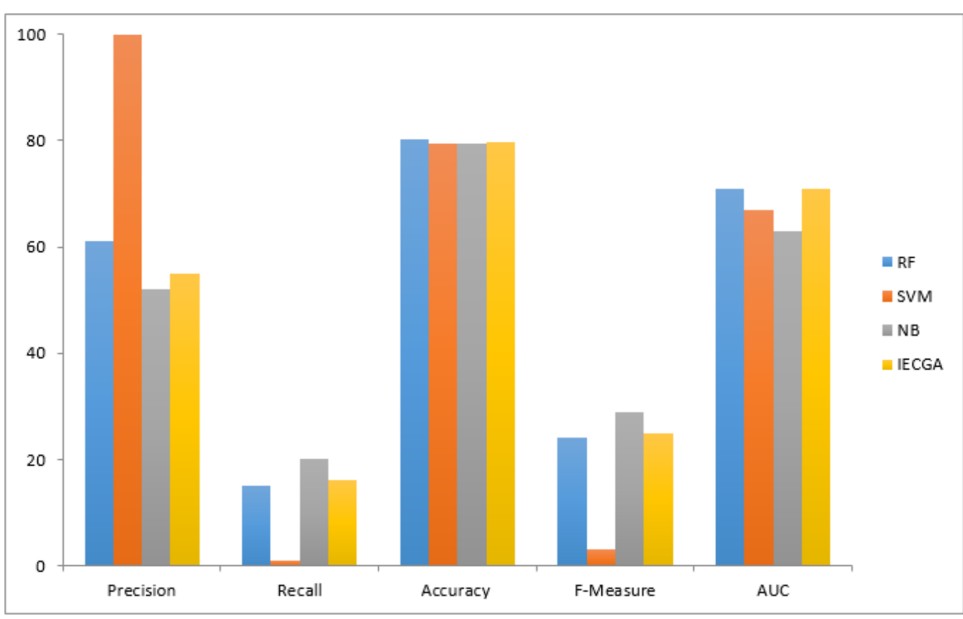

**Figure 8** **Performance measures on the JM1 testing dataset.**

precision and recall. The proposed IECGA framework demonstrated perfect precision, good recall, and high accuracy during training. In the testing phase, RF maintained a good balance between precision and recall with competitive accuracy. SVM showed low recall

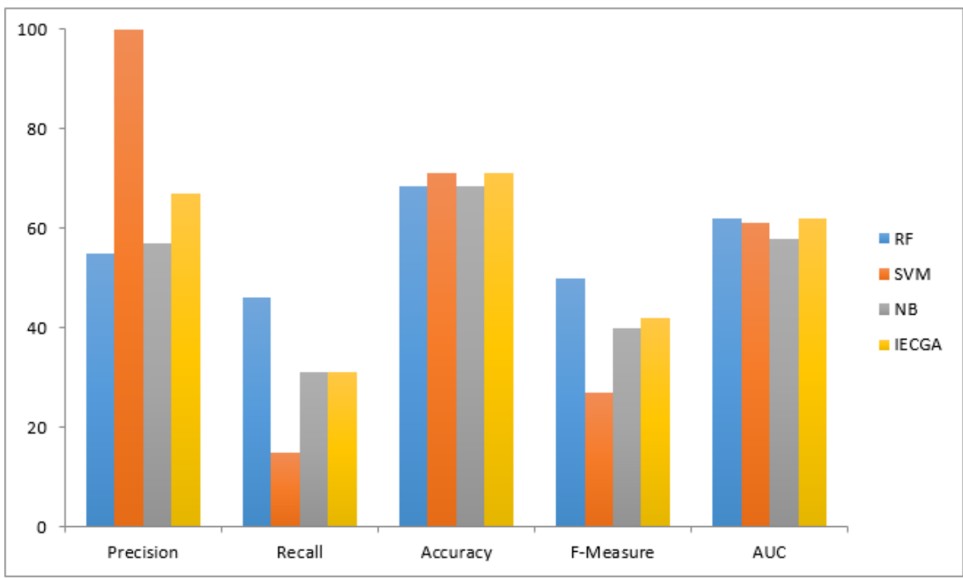

**Figure 9** Performance measures on the MC2 testing dataset.

during testing, while NB improved recall, resulting in a good F-measure and accuracy. The proposed IECGA framework reasonably balanced precision and recall during testing.

The graphical representation of all performance measures on the MW1 testing dataset is shown in Fig. 10.

Results achieved from the PC1 dataset are presented in Table 9. It demonstrates that in training, RF achieved perfect precision and recall, while SVM had perfect precision but lower recall, and NB showed a balanced performance. The proposed IECGA framework exhibited perfect precision and good recall. In testing, RF maintained a balance between precision and recall, SVM had a lower recall but reasonable precision, and NB improved its recall. The proposed IECGA framework maintained a balanced performance.

The graphical representation of all performance measures on the PC1 testing dataset is shown in Fig. 11.

Table 10 displays the results obtained from PC3. The training phase exhibited perfect results for the random forest (RF) classifier, raising concerns about overfitting. In testing, RF maintained decent precision and recall, suggesting effective defect identification. Support vector machine (SVM) had low recall in both phases, indicating difficulty in recognizing defects. Naive Bayes (NB) achieved balanced results suitable for software defect prediction. The proposed IECGA framework excelled in training and maintained effectiveness in testing, showing potential for enhancing software quality assurance.

The graphical representation of all performance measures on the PC3 testing dataset is shown in Fig. 12.
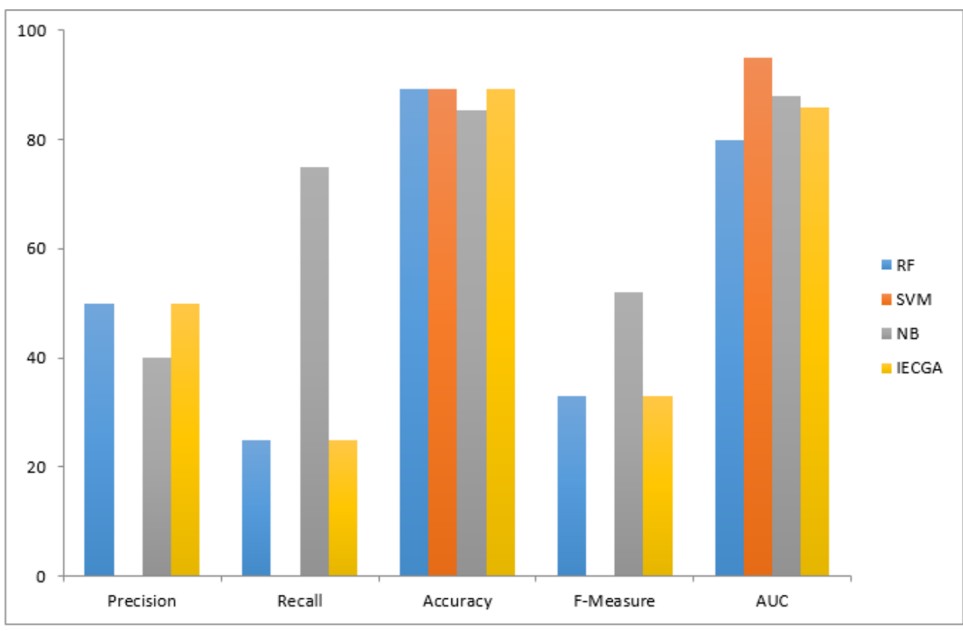

**Figure 10** Performance measures on the MW1 testing dataset.

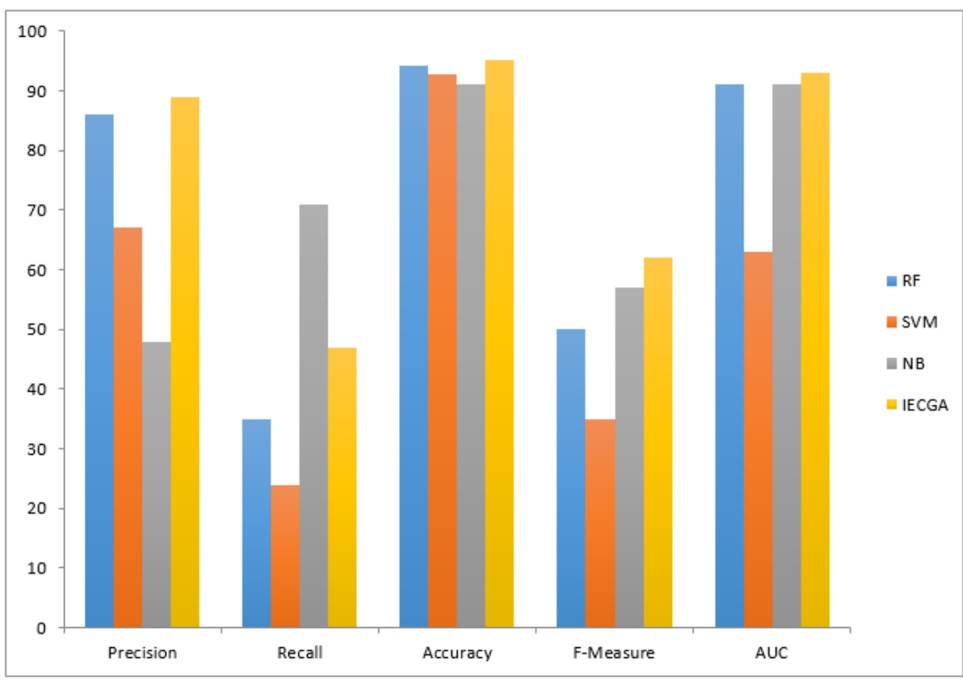

**Figure 11** Performance measures on the PC1 testing dataset.

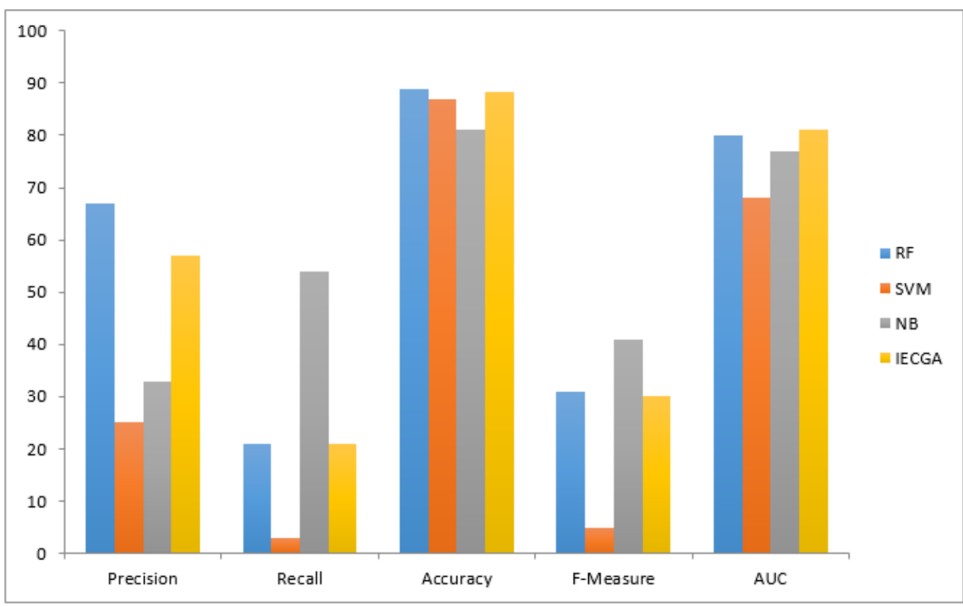

**Figure 12** **Performance measures on the PC3 testing dataset.**

Execution results on the PC4 dataset are displayed in Table 11. It represents that the RF classifier achieved perfection in the training phase, which may suggest over-fitting. However, in the testing phase, RF maintained decent precision and recall, demonstrating its ability to identify defective modules effectively. The SVM showed low recall in training and testing, indicating challenges in recognizing defects. NB achieved a balanced performance in both phases, making it suitable for software defect prediction. The proposed IECGA framework demonstrated impressive precision and competitive recall during training, and it maintained effectiveness in testing, showcasing its potential for enhancing software quality assurance.

The graphical representation of all performance measures on the PC4 testing dataset is shown in Fig. 13.

In the presented results, it is noteworthy that the RF classifier consistently exhibits overfitting across various datasets. This phenomenon can be attributed to the relatively small size of NASA datasets and their inherent class imbalance issues (*Liu et al., 2022*). The limited amount of data available for training can lead RF to excessively capture noise in the data, resulting in the observed overfitting behavior. At the same time, class imbalance may skew RF's predictions.

The IECGA framework delivers unbiased results by effectively integrating the accuracy of individual classifiers, ensuring a balanced and reliable approach to software defect prediction across various training datasets, as presented in Fig. 14.

IECGA framework provides unbiased results by effectively integrating the accuracy of individual classifiers, ensuring a well-balanced and dependable approach to software defect prediction across a range of testing datasets, as shown in Fig. 15.

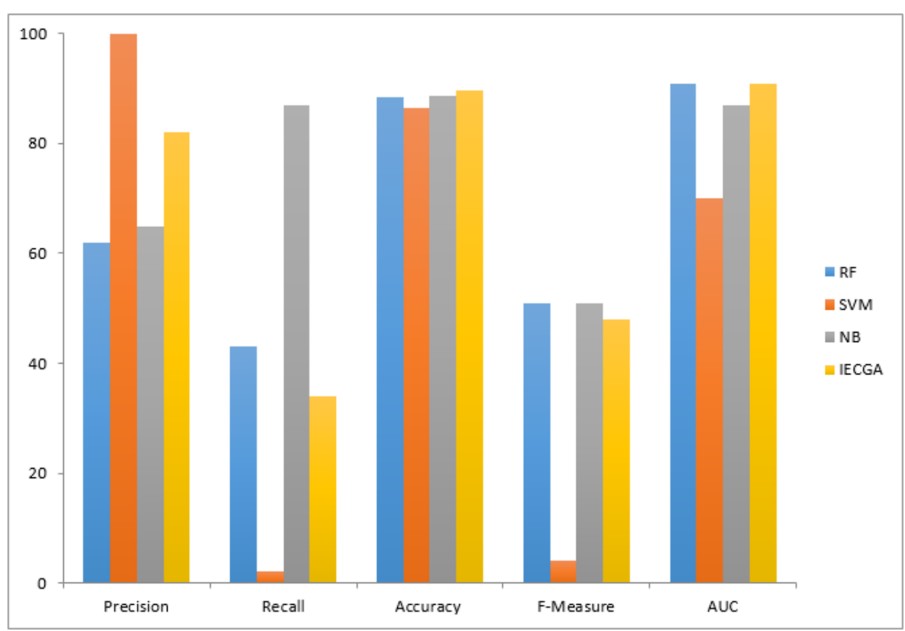

**Figure 13  Performance measures on the PC4 testing dataset.**

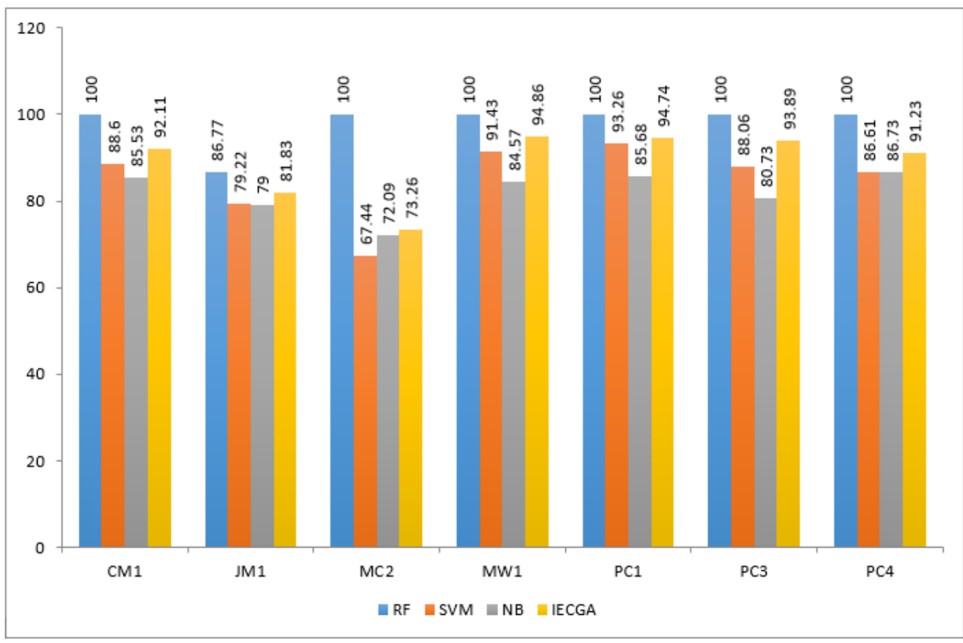

**Figure 14  Performance comparison based on classification accuracy of training data.**

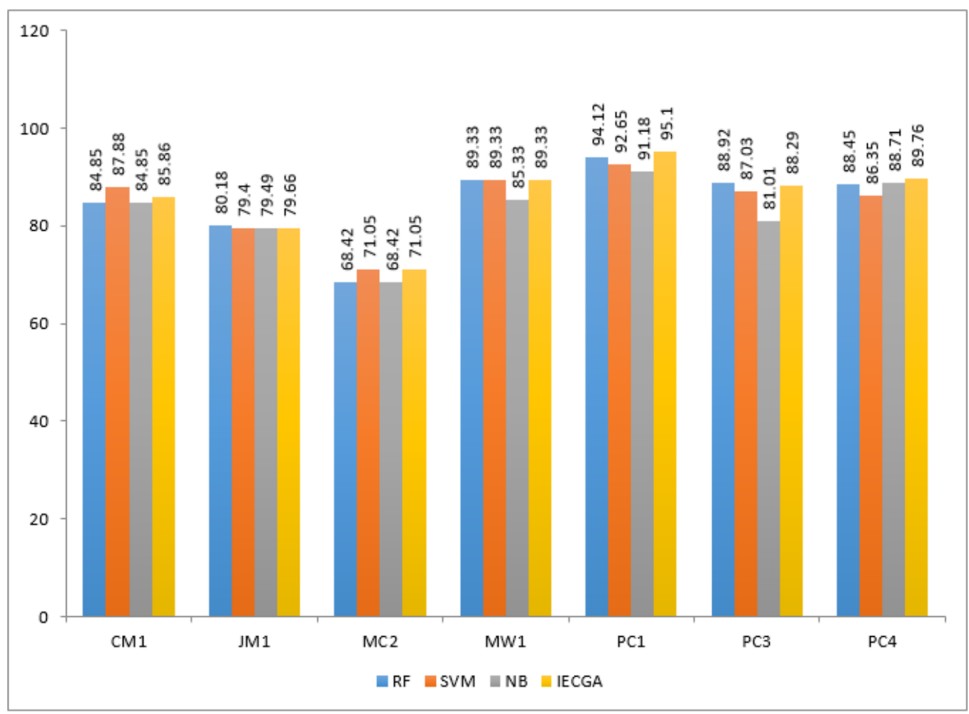

**Figure 15** Performance comparison based on classification accuracy of testing data.

**Table 12  GA-based accuracy comparison on training datasets.**

| Dataset | CM1 | JM1 | MC2 | MW1 | PC1 | PC3 | PC4 |
|---|---|---|---|---|---|---|---|
| No FS-Accuracy% | 91.67 | 81.31 | 74.42 | 95.43 | 94.95 | 98.78 | 88.64 |
| FS-Accuracy% | 92.11 | 81.83 | 73.26 | 94.86 | 94.74 | 93.89 | 91.23 |

## Genetic algorithm-based performance comparison

A comparative analysis has been conducted to evaluate the IECGA framework's accuracy with and without feature selection (FS). The evaluation encompasses training and testing datasets, highlighting the influence of the GA-based feature selection on the overall effectiveness of the framework. Table 12 presents the performance comparison on the training datasets, showing the impact of FS using genetic algorithms within the IECGA framework. Figure 16 shows the GA-based accuracy comparison on training datasets.

In the case of the CM1, JM1, and PC4 datasets, the FS led to a significant increase in accuracy, from 91.67% to 92.11%, 81.31% to 81.83%, and 88.64% to 91.23%, respectively. This reflects the effectiveness of the GA-based FS in identifying and retaining the most informative features, potentially reducing computational costs. For MC2, MW1, PC1, and PC3 datasets, there is a marginal decrease in accuracy with FS from 74.42% to 73.26%, 95.43% to 94.86%, 94.95% to 94.74%, and 98.78% to 93.89%. While there is a slight decrease in accuracy, the reduced number of features contributes to significant savings in computational resources, including time, money, and manpower. This emphasizes

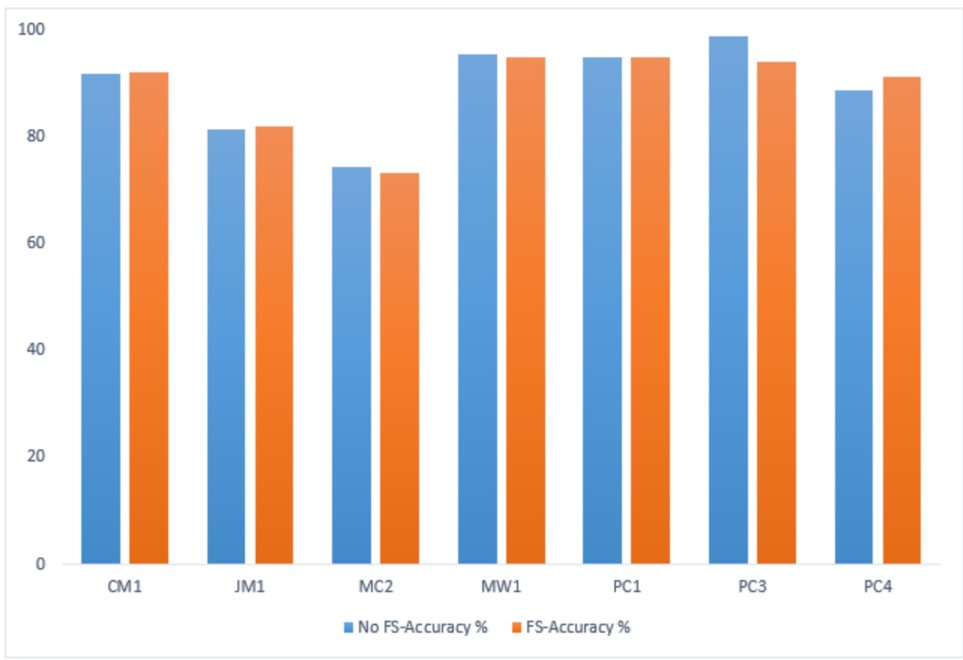

**Figure 16  GA-based accuracy comparison on training datasets.**

the trade-off between feature richness and the practical efficiency of the classifiers in real-world applications. Figure 17 shows the graphical representation of the GA-based accuracy comparison on training datasets.

Table 13 presents the performance comparison on the testing datasets, showing the impact of FS using genetic algorithms within the IECGA framework.

In the MC2, PC1, and PC4 datasets, FS contributes to an accuracy boost from 68.42% to 71.05%, 93.63% to 95.1%, and 87.66% to 89.76%, respectively. The results indicate that the GA effectively selects and retains pertinent features, optimizing the predictive performance of the IECGA framework. Despite the accuracy of 85.86% in CM1 and 89.33% in MW1 datasets, the GA-based feature selection is influential in reducing computational resources. The maintained accuracy indicates that the GA accurately identifies and preserves the most relevant features. The marginal reduction in accuracy is observed in the JM1 and PC3 datasets, from 79.84% to 79.66% and 88.92% to 88.29%, respectively. The trade-off between accuracy and resource optimization is evident, emphasizing FS's strategic application in scenarios where resource efficiency's benefits outweigh the marginal decrease in accuracy. Figure 17 shows the graphical representation of the GA-based accuracy comparison on testing datasets.

While GA-based FS aims to retain the most relevant features, its intrinsic evolutionary nature may occasionally lead to the omission of some features that, although less impactful individually, contribute collectively to the framework's predictive ability (*Hamdia, Zhuang & Rabczuk, 2021*; *Katoch, Chauhan & Kumar, 2021*). This accuracy drop should not be viewed solely as a compromise; instead, it reflects a strategic decision to prioritize

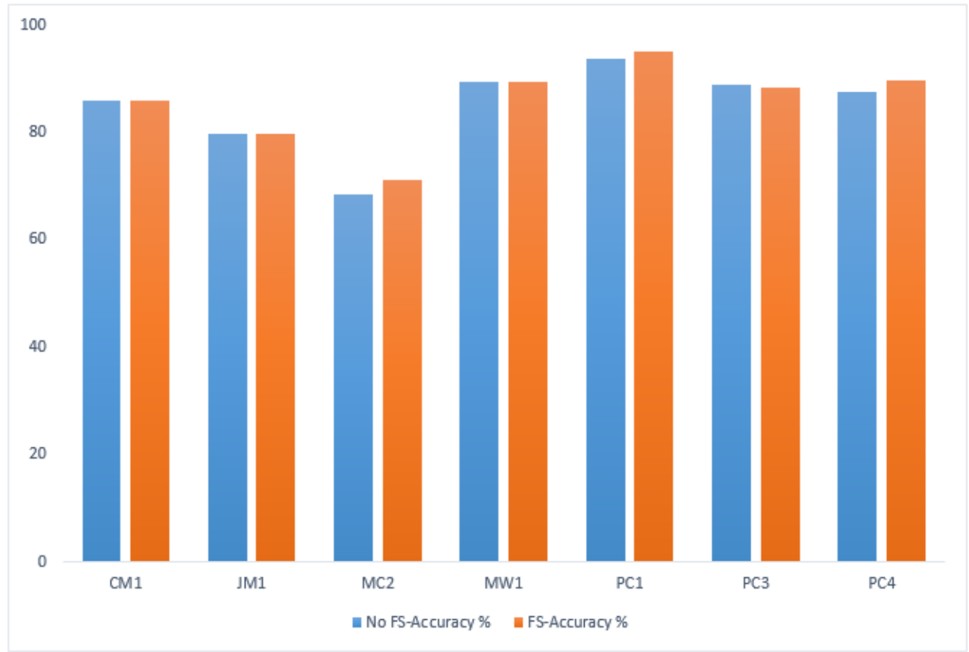

**Figure 17** GA-based accuracy comparison on testing datasets.

**Table 13** GA-based accuracy comparison on testing datasets.

| Dataset | CM1 | JM1 | MC2 | MW1 | PC1 | PC3 | PC4 |
|---|---|---|---|---|---|---|---|
| No FS-Accuracy% | 85.86 | 79.84 | 68.42 | 89.33 | 93.63 | 88.92 | 87.66 |
| FS-Accuracy% | 85.86 | 79.66 | 71.05 | 89.33 | 95.1 | 88.29 | 89.76 |

computational efficiency and interpretability by discarding features that might introduce noise or redundancy.

## Computational efficiency evaluation

Investigating the execution time of the framework with and without feature selection reveals notable improvements in computational efficiency. The application of FS consistently resulted in reduced execution time in both training and testing phases across all datasets. A comparative analysis of the execution time on both training and testing datasets has been presented in Table 14. Specifically, the time spent training with the original features significantly decreases when FS is employed, presenting a streamlined computational process. Similarly, in the testing phase, FS demonstrates its effectiveness by substantially lowering execution time, indicative of its role in optimizing the framework's overall computational efficiency. Notably, the IECGA framework demonstrates enhanced efficiency, achieving a substantial average reduction of 51.52% in training times and 52.31% in testing times. These findings emphasize the practical advantages of incorporating FS, showcasing its potential to enhance computational performance.

**Table 14 Comparative analysis of the execution time on training and testing datasets.**

| Dataset | Sample | Execution time without FS (seconds) | Execution time with FS (seconds) | Reduction in time (%) |
|---------|--------|-------------------------------------|----------------------------------|------------------------|
| CM1 | Training | 6.061 | 2.322 | 61.684 |
|     | Testing | 4.908 | 2.174 | 55.697 |
| JM1 | Training | 86.555 | 85.161 | 1.611 |
|     | Testing | 34.248 | 32.417 | 5.346 |
| MC2 | Training | 2.659 | 1.677 | 36.93 |
|     | Testing | 2.522 | 1.633 | 35.266 |
| MW1 | Training | 3.749 | 1.891 | 49.546 |
|     | Testing | 3.570 | 1.775 | 50.267 |
| PC1 | Training | 7.315 | 2.815 | 61.520 |
|     | Testing | 7.302 | 2.669 | 63.449 |
| PC3 | Training | 18.902 | 5.749 | 69.585 |
|     | Testing | 17.831 | 6.0458 | 66.094 |
| PC4 | Training | 31.958 | 6.466 | 79.767 |
|     | Testing | 32.587 | 3.234 | 90.073 |

**Figure 18 Comparative analysis of the execution time on training and testing datasets.**

A graphical representation of the comparative analysis of the execution time on both training and testing datasets has been reflected in Fig. 18.

## Performance comparison

In this section, the accuracy of the proposed feature selection-based ensemble software defect prediction (IECGA) framework is compared to state-of-the-art software defect

**Table 15  Accuracy comparison of the IECGA framework with modern techniques.**

| Datasets | CM1 | JM1 | MC2 | MW1 | PC1 | PC3 | PC4 |
|---|---|---|---|---|---|---|---|
| *Alsaeedi & Khan (2019)* | 83 | 78 | 68 | | 91 | 84 | 84 |
| *Balogun et al. (2019a)* | 84.1 | | | 83.6 | | | |
| *Aljamaan & Alazba (2020)* | 82.28 | 78.08 | 68.59 | 86.52 | 91.15 | 85.79 | |
| *Iqbal et al. (2019a)* | 77.55 | 73.96 | 64.86 | 82.66 | 92.64 | 82.59 | 86.08 |
| *Azam, Nouman & Gill (2022)* | 83 | 77 | | | 93.86 | | |
| *Mehta & Patnaik (2021)* | 84 | | | | 93.6 | | |
| *Balogun et al. (2020a)* | 83.79 | | 69.6 | 83.79 | | | |
| *Goyal & Bhatia (2020)* | 84.94 | 77.93 | | | | | |
| *Bhutamapuram & Sadam (2022)* | | 31.79 | | 83.02 | | | |
| *Iqbal & Aftab (2020)* | 79.59 | 62.78 | 62.16 | 77.33 | 89.65 | 75.94 | 74.8 |
| *Goyal (2022)* | | 49.43 | | | 92.99 | | |
| *Goyal (2020)* | | | | | 93 | | |
| *Alkhasawneh (2022)* | | | 67.29 | 86.89 | | | |
| *Balogun et al. (2020b)* | | | 70.16 | | | | 86.61 |
| *Iqbal et al. (2019b)* | | | 67.56 | | 89.7 | 87.34 | – |
| *Balogun et al. (2019b)* | | | 68.32 | 60.05 | 94.96 | 73.7 | 86.9 |
| *Alsghaier & Akour (2020)* | | | 67.09 | | 93.66 | | 88.2 |
| *Javed (2021)* | | | | | 93.59 | | |
| *Shafiq et al. (2023)* | | | | | 92 | | |
| *Balogun et al. (2021)* | | | | | 91.9 | 84.59 | 88.89 |
| *Singh & Haider (2022)* | | | | | 92.4 | | |
| *Amin (2019)* | | | | | 94 | | |
| *Cetiner & Sahingoz (2020)* | | | | | 92.2 | | |
| *Bajeh et al. (2022)* | | | | | | 81.92 | |
| *Mumtaz et al. (2021)* | 81.79 | | | | 89.79 | | |
| IECGA | 85.86 | 79.66 | 71.05 | 89.33 | 95.1 | 88.29 | 89.76 |

prediction techniques, as implemented in recent research conducted over the past five years. The comparison involves twenty published studies, with a focus on various datasets. Specifically, CM1 is the subject of ten studies; eight researchers examined JM1, MC2 was investigated in ten studies, MW1 was explored in eight studies, and PC1 was experimented with in eighteen studies. Additionally, the PC3 was analyzed in eight studies, and the PC4 was examined in seven. The results highlight the significance of incorporating heterogeneous base classifiers into feature selection-based ensemble classification, as it substantially enhances the accuracy of the software defect prediction process. The accuracy comparison between the IECGA framework and modern techniques is detailed in Table 15.

The graphical representation of the accuracy comparison of the IECGA framework with that of state-of-the-art techniques is shown in Fig. 19.

## THREATS TO VALIDITY

Validity threats encompass elements or challenges that could diminish research results' precision, reliability, or applicability. These challenges can potentially emerge at

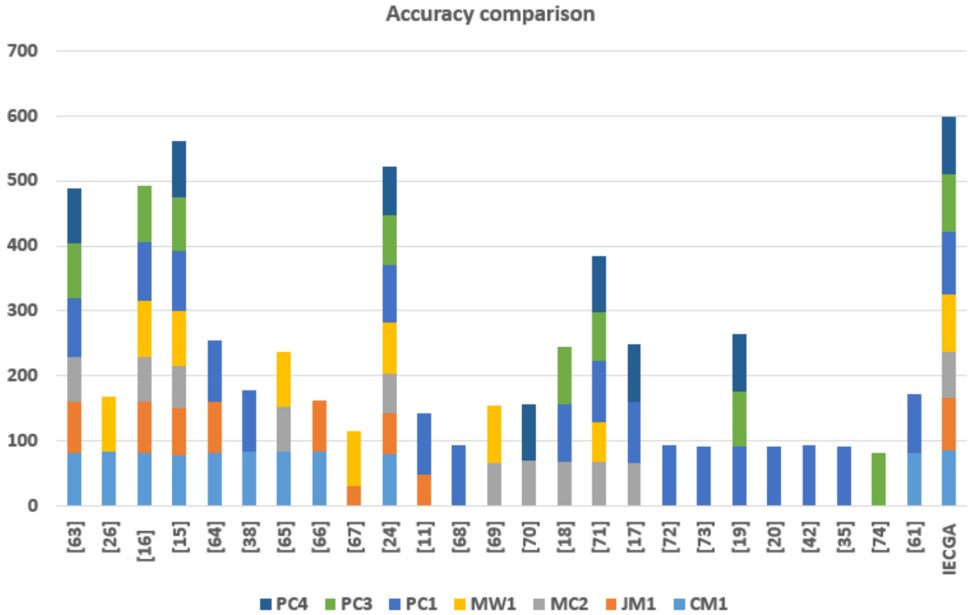

**Figure 19** Accuracy comparison of the IECGA framework with modern techniques.

various phases of the research endeavor and may impact the integrity of the study's deductions (*Yucalar et al., 2020*). Here are some of the paramount validity threats to consider:

**Internal validity**: It is related to evaluating whether the selected prediction techniques are suitable for the particular datasets utilized in this study or for other datasets addressing similar issues (*Sharma B & Sadam, 2022*). In this investigation, we employed three supervised classification algorithms—RF, SVM, and NB along with GA for feature selection—each characterized by distinct computational mechanisms and performance. However, GA may have limitations, such as sensitivity to parameter tuning and the potential for suboptimal convergence. While the Genetic Algorithm has demonstrated effectiveness, future research could explore alternative feature selection methods such as recursive feature elimination (RFE), information gain, principal component analysis (PCA), or hybrid approaches combining multiple techniques further to enhance the internal validity of software defect prediction frameworks. Additionally, researchers may consider incorporating clustering algorithms in conjunction with feature selection techniques to assess their impact on the efficiency of software defect prediction frameworks.

**External validity**: It examines whether the proposed solution is equally effective when applied to other datasets associated with the same problem domain (*Abdu et al., 2022*). This research employed seven benchmark datasets, namely CM1, JM1, MC2, MW1, PC1, PC3, and PC4 from NASA's defect repository, to implement the proposed IECA framework. Hence, the conclusion of this research cannot be generalized to other defect datasets having different attributes. However, the pre-processing steps, including dataset splitting, feature

selection, and parameter optimization in the classification step, can be implemented by other researchers in their studies.

**Construct validity**: This is associated with the suitability of the chosen performance measures for assessing the proposed frameworks' performance (*Liu et al., 2022*). In this research, five performance metrics, precision, recall, accuracy, F-measure, and AUC, were employed to evaluate the effectiveness of the proposed IECGA framework. Nevertheless, only the accuracy metric was utilized to compare the performance with state-of-the-art techniques.

**Conclusion validity**: It is relevant to the degree to which the inferences made in a study faithfully reflect the genuine relationships or effects observed in the proposed model (*Abdu et al., 2022*). In this research, the inference is derived from the accuracy comparison with state-of-the-art techniques, demonstrating better results of the proposed framework than contemporary research.

# CONCLUSION

This research presents an intelligent feature selection-based voting ensemble software defect prediction formwork named IECGA, which provides a promising approach to enhancing the accuracy and effectiveness of the software defect prediction process. Through extensive experimentation on NASA datasets, including CM1, JM1, MC2, MW1, PC1, PC3, and PC4, the IECGA framework demonstrated its ability to achieve competitive accuracy consistently. Its capacity to mitigate class imbalance issues and provide unbiased results is particularly noteworthy, making it a valuable tool in software quality assurance. Integrating genetic algorithms for feature selection further bolstered the model's predictive power. A comprehensive comparison with contemporary state-of-the-art techniques, involving twenty-five recent studies, highlighted the prowess of IECGA in accuracy across diverse datasets. These findings underscore the model's potential to contribute significantly to early defect detection in software development, ultimately creating more reliable and high-quality software products. As the software industry continues to evolve, the IECGA framework is a promising pre-testing solution to meet the ongoing challenge of software defect prediction. In the future, the focus should be on enhancing defect prediction models' identification and pattern recognition capabilities, possibly through advanced feature extraction techniques, deep learning, and transfer learning.

## Funding

This work was funded by the Princess Nourah bint Abdulrahman University Researchers Supporting Project number (PNURSP2024R235), Princess Nourah bint Abdulrahman University, Riyadh, Saudi Arabia. The funders had a role in study design, final proofreading and final preparation of the manuscript, and decision to publish. The funders had no role in data collection and analysis.

## Grant Disclosures

The following grant information was disclosed by the authors:

Princess Nourah bint Abdulrahman University Researchers Supporting Project number: PNURSP2024R235.

Princess Nourah bint Abdulrahman University, Riyadh, Saudi Arabia.

## Competing Interests

The authors declare there are no competing interests.

## Author Contributions

- Misbah Ali conceived and designed the experiments, performed the experiments, analyzed the data, performed the computation work, prepared figures and/or tables, authored or reviewed drafts of the article, writing code, and approved the final draft.
- Tehseen Mazhar conceived and designed the experiments, performed the experiments, analyzed the data, performed the computation work, prepared figures and/or tables, authored or reviewed drafts of the article, writing draft,Supervision, and approved the final draft.
- Amal Al-Rasheed conceived and designed the experiments, performed the experiments, analyzed the data, performed the computation work, prepared figures and/or tables, authored or reviewed drafts of the article, proof reading, and approved the final draft.
- Tariq Shahzad conceived and designed the experiments, performed the experiments, analyzed the data, performed the computation work, prepared figures and/or tables, authored or reviewed drafts of the article, data Collection, and approved the final draft.
- Yazeed Yasin Ghadi conceived and designed the experiments, performed the experiments, analyzed the data, performed the computation work, prepared figures and/or tables, authored or reviewed drafts of the article, proof reading,Investigation, and approved the final draft.
- Muhammad Amir Khan conceived and designed the experiments, performed the experiments, analyzed the data, performed the computation work, prepared figures and/or tables, authored or reviewed drafts of the article, edit and proof reading,adminstration, and approved the final draft.

## Data Availability

The raw data is available in the Supplemental Files.

## Supplemental Information

Supplemental information for this article can be found online at http://dx.doi.org/10.7717/peerj-cs.1860#supplemental-information.

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
