# Peer review of "Enhancing software defect prediction: a framework with improved feature selection and ensemble machine learning"

_PeerJ Computer Science, doi:10.7717/peerj-cs.1860_

## Round 0.1 · original submission · Major Revisions

Dear authors,

Thank you for submitting your article. The reviewers' comments are now available. Your article has not been recommended for publication in its current form. However, we encourage you to address the reviewers' concerns and criticisms; particularly regarding readability, quality, experimental design and validity, and resubmit your article once you have updated it accordingly.

Reviewer 2 and Reviewer 3 have asked you to provide specific references. You are welcome to add them if you think they are relevant. However, you are not required to include these citations, and if you do not, it will not affect my decision.

When submitting the revised version of your article, it will be better to address the following:

1- The research gaps and contributions should be clearly summarized in the introduction section. Please evaluate how your study is different from others in the related section.

2- The values for the parameters of the algorithms selected for comparison are not given. The analysis and configurations of experiments should be presented in detail for reproducibility. A table with parameter settings for experimental results and analysis should be included in order to clearly describe them.

3- The paper lacks the running environment, including software and hardware. The analysis and configurations of experiments should be presented in detail for reproducibility. It is convenient for other researchers to redo your experiments and this makes your work easy acceptance.

4- Pros and cons of the method should be clarified. What are the research limitation(s) methodology(ies) adopted in this work?

5- English grammar and scientific writing style errors should be corrected.

6- Equations should be correctly written accortding to a single style. Explanations of the equations and variables used in these equations should also be checked. All variables should be written in italic as in the equations.

7- The paper should correctly and appropriately present how computational complexity is reduced, while preserving the essential information needed to make accurate predictions.

**Language Note:** The Academic Editor has identified that the English language must be improved. PeerJ can provide language editing services - please contact us at copyediting@peerj.com for pricing (be sure to provide your manuscript number and title). Alternatively, you should make your own arrangements to improve the language quality and provide details in your response letter. – PeerJ Staff

Reviewer 1 ·

Basic reporting

1-First of all, the situation I need to point out is this: Figure 10 and Figure 11 shows the results of some evaluation metrics for the same PC1 data set. I think the authors mistakenly wrote PC1 for the PC3 data set. This needs to change.
2- Another problem is that instead of using variables such as alpha and beta in equations 4, 5, 6, and 7 used to calculate the evaluation metrics, TP, TN, FP, and FN variables should be used.
3- The article talks about feature selection with the help of a Genetic algorithm. However, some special parameters of the genetic algorithm were not mentioned during the selection of these features. In other words, while the experiments are being carried out, I cannot see any information about the mutation probability, crossover probability, and population, which are the main parameters of the genetic algorithm.
4- Also, another question is why was the genetic algorithm chosen when there were hundreds of meta-heuristic optimization methods for this problem type? Also, why weren't other meta-heuristic algorithms used and their results compared?
5- The author mentions that the computational complexity has been significantly reduced. ("In the second accurate stage, a feature selection technique based on the Genetic Algorithm is applied to identify the optimal subset of features. This step significantly reduces computational complexity while preserving the essential information required for predictions."). However, when the entire article was reviewed, no data was found indicating that computational complexity had decreased. Authors must find the computational complexity of the methods they use and prove in the paper that it is reduced.

Experimental design

6- More information and visuals need to be provided about the data set used and the examples contained in these data sets.
7- The number of classes in the data sets used during the experiments is very small. It is not very appropriate to perform "software defect detection" on a data set containing 2 classes and make a generalization based on the results obtained. During training, RF classifier Precision, Recall, F_Measure, and AUC metric values were all 1 in the CM1, MC2, MW1, PC1, PC3 and PC4 data sets. In addition, the accuracy values were 100% in all of these data sets. This is not a normal situation.

Validity of the findings

8-As shown in Table 3, if we talk about the CM1 data set, it is extremely important how the 8 features selected among the 38 original features were obtained. This situation needs to be explained.
9- It should be noted that if the genetic algorithm was not used at all and all of the original features shown in Table 3 were sent to ML classifiers, would an extremely different situation arise in terms of evaluation metrics? In other words, the authors must give all the original features to the classifier and get the experimental results without making a feature selection. These results should also be compared.

Reviewer 2 ·

Basic reporting

The authors propose a comprehensive, five-stage framework for software defect prediction. The first stage involves selecting a cleaned version of NASA’s defect datasets, ensuring the integrity of the data. In the second stage, a feature selection technique based on the genetic algorithm is applied to identify the optimal subset of features. This step significantly reduces computational complexity while preserving the essential information required for accurate predictions. In the third step, three different binary classifiers are used as base classifiers. These are Random Forest, Support Vector Machine, and Naïve Bayes. Through iterative tuning, the classifiers are optimized to achieve the highest level of accuracy individually. In the fourth stage, an ensemble machine-learning technique known as voting is applied as a master classifier, leveraging the collective decision-making power of the base classifiers. This ensemble approach enhances the overall prediction accuracy and robustness of the framework. The final stage evaluates the performance of the proposed framework using five widely recognized performance evaluation measures: precision, recall, accuracy, F-measure, and area under the curve. These measures provide a comprehensive analysis of the framework’s effectiveness in predicting software defects. However, it is necessary to improve the following points:
• Improving the abstract's structure would involve incorporating distinct sections for the study's background, objectives, materials and methods, results, conclusions, and recommendations. Currently, the abstract resembles an introductory background statement, which should be refined to make it more quantitative in nature. Additionally, it is advisable to reduce the extent of the background introduction in the abstract.
• Figures 1, and 2 should be cited
• The authors should better explain some materials and methods pointed out in this section. For example: (i) Genetic Algorithm (ii) Random Forest, (iii) Support Vector Machine, (iv) and Naïve Bayes. Also, provide references to these methods. Thus, it is necessary to present theoretical and practical concepts of the implementation of these machine learning methods.
• Authors should dedicate a new section to reviewing related works, identifying gaps observed in the reviewed studies, and explaining how their work aims to address the limitations identified in the existing literature.
• The study's limitations should be explicitly acknowledged, and suggestions for future research directions should be presented.
• Ensuring replicability, authors should include the source codes required to reproduce the study's results.
• Overall, there is a need for substantial improvement in both the English language usage and the presentation style. Numerous grammatical errors and typos were noted. It is recommended that the authors seek the assistance of a colleague proficient in English and well-versed in the subject matter or consider employing professional editing services.
• The authors are encouraged to incorporate more recent references from 2022 and 2023 into their work. To facilitate this, I have provided a list of publications relevant to their study, which I recommend they cite and reference in their paper. These recent articles can significantly enhance the quality of the manuscript and contribute to the overall value of the journal:


a. Yang, S., Li, Q., Li, W., Li, X., & Liu, A. (2022). Dual-Level Representation Enhancement on Characteristic and Context for Image-Text Retrieval. IEEE Transactions on Circuits and Systems for Video Technology, 32(11), 8037-8050. doi: 10.1109/TCSVT.2022.3182426
b. Zhou, X., & Zhang, L. (2022). SA-FPN: An effective feature pyramid network for crowded human detection. Applied Intelligence, 52(11), 12556-12568. doi: 10.1007/s10489-021-03121-8
c. Long, W., Xiao, Z., Wang, D., Jiang, H., Chen, J., Li, Y.,... Alazab, M. (2023). Unified Spatial-Temporal Neighbor Attention Network for Dynamic Traffic Prediction. IEEE Transactions on Vehicular Technology, 72(2), 1515-1529. doi: 10.1109/TVT.2022.3209242
d. Ding, Y., Zhang, W., Zhou, X., Liao, Q., Luo, Q.,... Ni, L. M. (2021). FraudTrip: Taxi Fraudulent Trip Detection From Corresponding Trajectories. IEEE Internet of Things Journal, 8(16), 12505-12517. doi: 10.1109/JIOT.2020.3019398
e. Hou, X., Zhang, L., Su, Y., Gao, G., Liu, Y., Na, Z.,... Chen, T. (2023). A space crawling robotic bio-paw (SCRBP) enabled by triboelectric sensors for surface identification. Nano Energy, 105, 108013. doi: 10.1016/j.nanoen.2022.108013
f. Luo, J., Wang, Y., & Li, G. (2023). The innovation effect of administrative hierarchy on intercity connection: The machine learning of twin cities. Journal of Innovation & Knowledge, 8(1), 100293. doi: https://doi.org/10.1016/j.jik.2022.100293
g. Luo, J., Wang, G., Li, G., & Pesce, G. (2022). Transport infrastructure connectivity and conflict resolution: a machine learning analysis. Neural Computing and Applications, 34(9), 6585-6601. doi: 10.1007/s00521-021-06015-5
h. Xu, J., Guo, K., Zhang, X., & Sun, P. Z. H. (2023). Left Gaze Bias between LHT and RHT: A Recommendation Strategy to Mitigate Human Errors in Left- and Right-Hand Driving. IEEE Transactions on Intelligent Vehicles. doi: 10.1109/TIV.2023.3298481
i. Chen, J., Xu, M., Xu, W., Li, D., Peng, W.,... Xu, H. (2023). A Flow Feedback Traffic Prediction Based on Visual Quantified Features. IEEE Transactions on Intelligent Transportation Systems, 24(9), 10067-10075. doi: 10.1109/TITS.2023.3269794.

Experimental design

• The authors are encouraged to detail the methodology used to assess or evaluate the study's performance.
• The authors must provide a detailed description of the dataset used for the implementation process. The ratio of the training set, validation set, and testing set should be presented.
• How did the authors tune the optimal hyperparameter of all models? It should be described clearly.
• It is essential to compare the study with contemporary systems considered state-of-the-art, elucidating how it surpasses these systems and providing insights into any performance disparities.
• The general quality of figures is not so good, so you should update them with higher resolution

Validity of the findings

No comment

Additional comments

No comment

·

Basic reporting

The primary contribution of this research lies in the preprocessing step, precisely the feature selection technique. By extracting the most relevant features, the proposed framework achieves enhanced accuracy, reduces computational costs, and improves interpretability compared to existing methods.

• The abstract would benefit from a revised structure that includes sections for the study's background, objectives, materials and methods, results, conclusions, and recommendations. It appears more like an introductory background statement that the authors should let the abstract be quantitative. The background introduction of the abstract should also be decreased.
• The research issue and objectives should be highlighted in a separate paragraph in the introduction section (second to the last paragraph)
• Since the work uses models like Random Forest, Support Vector Machine, and Naïve Bayes, then the motives of using these models should be explained, and what is your combination logic for them?
• In recent years, there have emerged many efficient software defect predictions, and some recent ones should be discussed and compared.
• In line 163, the “MATERIALS AND METHODS” should be consistent with others by changing it to sentence case
• Three classification methods were tested as baselines. Did the authors include the widely used gradient boosting method, such as XGBoost, for comparison? Typically, XGBoost performs well in similar tasks.
• The overall quality of the figures is somewhat lacking; therefore, it is recommended that you enhance them by using higher resolution images.
• The language quality should be enhanced, especially the tense issue. The quality of the English language and presentation needs significant improvement. There were many typos and grammar issues in the text. To ensure your paper is of high quality, it's advisable to seek assistance from a colleague proficient in English and knowledgeable in the subject matter. Alternatively, consider hiring a professional editing service for their expertise.
• The authors should show more recent references from the last two years (2022 and 2023). In light of this, I have suggested publications relating to your study that I want the authors to cite and reference in their papers. These are recent articles and will be of benefit to the author's manuscript and the journal as well:
a. Liu, X., Zhou, G., Kong, M., Yin, Z., Li, X., Yin, L.,... Zheng, W. (2023). Developing Multi-Labelled Corpus of Twitter Short Texts: A Semi-Automatic Method. Systems, 11(8), 390. doi: 10.3390/systems11080390
b. Li, D., Ortegas, K. D., & White, M. (2023). Exploring the Computational Effects of Advanced Deep Neural Networks on Logical and Activity Learning for Enhanced Thinking Skills. Systems, 11(7), 319. doi: 10.3390/systems11070319
c. Liu, X., Wang, S., Lu, S., Yin, Z., Li, X., Yin, L.,... Zheng, W. (2023). Adapting Feature Selection Algorithms for the Classification of Chinese Texts. Systems, 11(9), 483. doi: 10.3390/systems11090483
d. Tao, Y., Shi, J., Guo, W., & Zheng, J. (2023). Convolutional Neural Network Based Defect Recognition Model for Phased Array Ultrasonic Testing Images of Electrofusion Joints. Journal of Pressure Vessel Technology, 145(2). doi: 10.1115/1.4056836
e. Liu, X., Shi, T., Zhou, G., Liu, M., Yin, Z., Yin, L.,... Zheng, W. (2023). Emotion classification for short texts: an improved multi-label method. Humanities and Social Sciences Communications, 10(1), 306. doi: 10.1057/s41599-023-01816-6
f. Zhang, R., Li, L., Zhang, Q., Zhang, J., Xu, L., Zhang, B.,... Wang, B. (2023). Differential Feature Awareness Network within Antagonistic Learning for Infrared-Visible Object Detection. IEEE Transactions on Circuits and Systems for Video Technology. doi: 10.1109/TCSVT.2023.3289142
g. Jiang, H., Wang, M., Zhao, P., Xiao, Z., & Dustdar, S. (2021). A Utility-Aware General Framework With Quantifiable Privacy Preservation for Destination Prediction in LBSs. IEEE/ACM Trans. Netw., 29(5), 2228-2241. doi: 10.1109/TNET.2021.3084251

Experimental design

No Comment

Validity of the findings

No Comment

Additional comments

No Comment

---

## Round 0.2 · Minor Revisions

Dear authors,

The reviewers have now commented on your revised manuscript. Although two reviewers are satisfied with the additions and changes, reviewer 1 says that his comments have not been fully addressed. Please consider the advice and comments of reviewer 1 and resubmit your manuscript.

Best wishes,

Reviewer 1 ·

Basic reporting

Since there is no change in my previous situation, I reject this article again.

Experimental design

Since there is no change in my previous situation, I reject this article again.

Validity of the findings

Since there is no change in my previous situation, I reject this article again.

Additional comments

Since there is no change in my previous situation, I reject this article again.

Reviewer 2 ·

Basic reporting

I am writing to express my gratitude for the considerable effort you have put into revising your manuscript. Your responsiveness to the feedback provided and the thoroughness of your revisions have significantly enhanced the quality of your submission.

You have diligently addressed each of the concerns and suggestions that I highlighted in my review. The improvements made in areas such as methodology, data analysis, literature review, etc., have notably enriched the clarity, depth, and rigor of your paper.

Consequently, I am pleased to inform you that I have recommended your manuscript for acceptance in PeerJ Journal. I believe that your work makes a valuable contribution and will be of interest to the readership of the paper.

Experimental design

No comment

Validity of the findings

No comment

Additional comments

No comment

·

Basic reporting

All raised issues has been addressed and the article is now suitable for recommendation for acceptance.

Experimental design

No comment

Validity of the findings

No comment

Additional comments

No comment

---

## Round 0.3 · accepted · Accept

Dear authors,

Thank you for the revision. The paper seems to be improved in the opinion of the reviewers. The paper is now ready to be published.

Best wishes,

Reviewer 1 ·

Basic reporting

.

Experimental design

.

Validity of the findings

.

Additional comments

The authors fulfilled all my other requests, especially the genetic algorithm parameters, during the revision phase. I congratulate the authors of this work, which I think will make a serious contribution to the world of science.